# Integrating deep learning CT-scan model, biological and clinical variables to predict severity of COVID-19 patients

Nathalie Lassau[1,2], Samy Ammari[1,2], Emilie Chouzenoux [3], Hugo Gortais[4], Paul Herent [5], Matthieu Devilder [4], Samer Soliman[4], Olivier Meyrignac [4], Marie-Pauline Talabard[4], Jean-Philippe Lamarque[1,2], Remy Dubois[5], Nicolas Loiseau[5], Paul Trichelair[5], Etienne Bendjebbar[5], Gabriel Garcia[1], Corinne Balleyguier[1,2], Mansouria Merad[6], Annabelle Stoclin[6], Simon Jegou[5], Franck Griscelli[7], Nicolas Tetelboum[1], Yingping Li[2,3], Sagar Verma [3], Matthieu Terris[3], Tasnim Dardouri[3], Kavya Gupta[3], Ana Neacsu[3], Frank Chemouni[6], Meriem Sefta[5], Paul Jehanno[5], Imad Bousaid[8], Yannick Boursin [8], Emmanuel Planchet[8], Mikael Azoulay[8], Jocelyn Dachary[5], Fabien Brulport[5], Adrian Gonzalez[5], Olivier Dehaene[5], Jean-Baptiste Schiratti[5], Kathryn Schutte[5], Jean-Christophe Pesquet[3], Hugues Talbot[3], Elodie Pronier[5], Gilles Wainrib[5], Thomas Clozel [5], Fabrice Barlesi[9], Marie-France Bellin[4] & Michael G. B. Blum [5]✉

The SARS-COV-2 pandemic has put pressure on intensive care units, so that identifying predictors of disease severity is a priority. We collect 58 clinical and biological variables, and chest CT scan data, from 1003 coronavirus-infected patients from two French hospitals. We train a deep learning model based on CT scans to predict severity. We then construct the multimodal AI-severity score that includes 5 clinical and biological variables (age, sex, oxygenation, urea, platelet) in addition to the deep learning model. We show that neural network analysis of CT-scans brings unique prognosis information, although it is correlated with other markers of severity (oxygenation, LDH, and CRP) explaining the measurable but limited 0.03 increase of AUC obtained when adding CT-scan information to clinical variables. Here, we show that when comparing AI-severity with 11 existing severity scores, we find significantly improved prognosis performance; AI-severity can therefore rapidly become a reference scoring approach.

[1] Imaging Department, Gustave Roussy, Université Paris -Saclay, Villejuif 94805, France. [2] Biomaps, UMR 1281 INSERM, CEA, CNRS, Université Paris-Saclay, Villejuif 94805, France. [3] Centre de Vision Numérique, Université Paris-Saclay, CentraleSupélec, Inria, 91190 Gif-sur-Yvette, France. [4] Radiology Department, Hôpital de Bicêtre – AP-HP, Université Paris-Saclay, Le Kremlin-Bicêtre, France. [5] Owkin Lab, Owkin, Inc, New York, NY, USA. [6] Département Interdisciplinaire d'Organisation des Parcours Patients, Gustave Roussy, Université Paris-Saclay, Villejuif 94805, France. [7] Département de Biologie, Gustave Roussy, Université Paris-Saclay, Villejuif 94805, France. [8] Direction de la Transformation Numérique et des Systèmes d'Information, Gustave Roussy, Université Paris-Saclay, 94805 Villejuif, France. [9] Département d'Oncologie Médicale, Gustave Roussy, Université Paris-Saclay, Villejuif 94805, France. ✉email: michael.blum@owkin.com

Hospitalized COVID-19 patients are likely to develop severe outcomes requiring mechanical ventilation or high-flow oxygenation. Among hospitalized patients, 14–30% will require admission to an intensive care unit (ICU), 12–33% will require mechanical ventilation, and 20–33% will die[1–4]. Detection at admission of patients at risk of severe outcomes is important to deliver proper care and to optimize use of limited ICU ressources[5].

Identification of hospitalized COVID-19 patients at risk for severe deterioration can be done using risk scores that combine several factors including age, sex, and comorbidities (CALL, COVID-GRAM, 4C Mortality Score)[6–12]. Some risk scores also include additional markers of severity such as the dyspnea symptom, clinical examination variables such as low oxygen saturation and elevated respiratory rate, as well as biological factors reflecting multi-organ failures such as elevated lactate dehydrogenase (LDH) values[8,10,13–15].

Beyond clinical and biological variables, computerized tomography (CT) scans also contain prognostic information, as the degree of pulmonary inflammation is associated with clinical symptoms, and the amount of lung abnormality is associated with severe evolution[16–20]. CT scans can be acquired at admission to diagnose COVID-19 when RT-PCR results are negative[21]. However, the extent to which CT scans at patient admission add prognostic information beyond what can be inferred from clinical and biological data is unresolved.

The objective of this study was to integrate clinical, biological, and radiological data to predict the outcome of hospitalized patients. By processing CT scan images with a deep learning model and by using a radiologist report that contains a semi-quantitative description of CT scans, we evaluated the additional amount of information brought by CT scans.

Here, we show that integrating clinical and biological data with a deep learning CT scan analysis more accurately predicts severity of COVID-19 among hospitalized patients than existing scores for severity.

## Results

A total of 1003 patients from Kremlin-Bicêtre (KB, Paris, France) and Gustave Roussy (IGR, Villejuif, France) were enrolled in the study. Clinical, biological, and CT scan images and reports were collected at hospital admission. There were 931 patients for whom clinical, biological, and CT scan data were available (Supplementary Fig. 1). A total of 506,341 images were analyzed for the 980 patients with available CT scans (average of 517 slices per scan). Radiologists annotated 17,873 images from 329 CT scans. Summary statistics for the clinical, biological, and CT scan data are provided in Table 1.

**Variables associated with severity**. We first evaluated how clinical and biological variables measured at admission were associated with future severe progression, which we defined as an oxygen flow rate of 15 L/min or higher and/or the need for mechanical ventilation and/or patient death[22]. This definition of severe progression corresponds to a score of 5 or more according to the World Health Organization evaluation of severity on a 1–10 scale. We computed the severity odds ratios for each individual variable, and at each hospital center (Table 1). When combining association results from the two centers, we found 12 variables significantly associated with severity ($p < 0.05/58$ to account for testing 58 variables, Table 1): age, sex, oxygen saturation, diastolic pressure, respiratory rate, chronic kidney disease, hypertension, LDH, and urea, CRP, polynuclear neutrophil, and leukocytes.

We then assessed the predictive value of features from admission radiology reports. These reports contain semi-quantitative evaluations of the extent of disease which values range from 0 to 5, as well as a presence/absence coding of several types of lung lesions in COVID-19 patients. We found three significant associated features ($p < 0.05/58$): extent of disease, and presence of crazy-paving lesions, which are both associated with greater severity, and presence of a peripheral distribution of lesions, which is associated with lesser severity.

**A neural network model to predict severity based on CT scans**. To capture CT scan prognosis information from images, we considered a weakly supervised approach with no radiologist-provided annotations (Supplementary Fig. 2)[23]. A deep learning model was trained to predict severe progression based on a CT scan image. The neural network was trained on a development cohort consisting of 646 patients from Kremlin-Bicêtre Hospital (KB). It was evaluated on 150 KB patients, who were leftover from the development cohort, and it was further evaluated using a validation cohort consisting of 135 patients from Institut Gustave Roussy hospital (IGR). The discriminative ability of the neural network was of AUC = 0.76 (0.67, 0.85) for the 150 leftover KB patients, who were not used to train the network, and of AUC = 0.75 (0.65,0.84) for the validation IGR dataset. As a point of comparison, the AUC obtained with the radiologist evaluation of disease extent is of 0.73 (0.64–0.82) for the 150 KB patients of the development cohort and of 0.66 (0.56–0.76) for the validation IGR cohort, and the difference between the two AUC values was significant for the validation IGR cohort only ($p \leq 0.05$).

**Interpretability analysis of the neural network model**. To apprehend the information present within the CT scans that is captured by the weakly supervised neural network model, we evaluated to what extent the features (internal representation) extracted by the neural network can predict clinical and radiological variables. To this end, we trained a new logistic regression with the extracted features as input, and some clinical and radiological variables as output. AUC on the 150 leftover patients of the KB development cohort was 0.93 (0.88,0.97) for disease extent (threshold > 2), 0.78 (0.70, 0.85) for crazy paving, 0.64 (0.53, 0.74) for condensation and 0.80 (0.65, 0.94) for ground glass opacity (GGO) (Supplementary Table 1). It was also possible to relate internal representations of the neural networks to clinical variables. We obtained an AUC of 0.88 (0.82, 0.94) for predicting an age strictly more than 60 years old, an AUC of 0.93 (0.89, 0.97) for sex, and of 0.76 (0.68, 0.84) for predicting an oxygen saturation more than 90%. As a comparison, a logistic regression trained on the variables from the radiology report obtained only AUC scores of 0.70 (0.61, 0.78) for age, 0.57 (0.48, 0.67) for sex, and of 0.68 (0.58, 0.77) for oxygen saturation, and differences of AUC were significant ($p < 0.05$). Simply put, this analysis shows that the internal representation of the neural network captures clinical features from the lung CTs, such as sex or age, on top of the known COVID-19 radiology features.

**A multimodal prognostic models for severity**. To add information from lab tests and chest characteristics to the CT scan information, we constructed the AI-severity score. We used a greedy search approach to include optimal clinical and biological variables (Methods). In addition to the CT deep learning variable, the variables included in AI-severity are age, sex, oxygen saturation, urea, and platelet counts. Coefficients and transformations required to compute the 6-variable AI-severity score are available in Supplementary Table 2. Coefficients required to compute AI-severity were learned using the WHO-defined high

**Table 1 Population description for the KB and IGR hospitals and association between variables measured at admission and severity.**

| | KB, N = 837 | | | IGR, N = 150 | | | Pooled | |
|---|---|---|---|---|---|---|---|---|
| | Distribution | Odds ratio | p-value association with severity | Distribution | Odds ratio | p-value association with severity | Pooled p-value | Significant association |
| **Outcomes** | | | | | | | | |
| Severity: "Intubation" OR "O$_2$ ≥ 15 L" OR "Death" | 30% [817] | – | – | 32% [144] | – | – | – | |
| Intubation | 13% [837] | – | – | 11% [150] | – | – | – | |
| O$_2$ ≥ 15 L | 15% [837] | – | – | 11% [150] | – | – | – | |
| Death | 17% [837] | – | – | 17% [150] | – | – | – | |
| **Clinical characteristics** | | | | | | | | |
| Age | 63.0 (52.0, 77.0) [833] | 1.61 (1.37, 1.90) | $7.32 \times 10^{-09}$ | 61.0 (49.0, 71.0) [149] | 1.35 (0.92, 1.98) | $1.29 \times 10^{-01}$ | $2.54 \times 10^{-09}$ | Yes |
| Sex | 57% [837] | 1.97 (1.43, 2.72) | $3.03 \times 10^{-05}$ | 49% [150] | 1.07 (0.51, 2.23) | $8.67 \times 10^{-01}$ | $3.52 \times 10^{-05}$ | Yes |
| BMI | 27.0 (23.5, 31.1) [400] | 1.17 (0.95, 1.43) | $1.39 \times 10^{-01}$ | 24.9 (21.5, 27.7) [108] | 1.28 (0.81, 2.00) | $2.87 \times 10^{-01}$ | $1.00 \times 10^{-01}$ | |
| Height | 1.7 (1.6, 1.8) [404] | 1.18 (0.96, 1.46) | $1.13 \times 10^{-01}$ | 1.7 (1.6, 1.8) [109] | 1.08 (0.69, 1.71) | $7.30 \times 10^{-01}$ | $1.05 \times 10^{-01}$ | |
| Weight | 75.0 (62.0, 90.0) [543] | 1.08 (0.91, 1.29) | $3.84 \times 10^{-01}$ | 72.0 (60.0, 85.0) [119] | 1.31 (0.86, 1.99) | $2.06 \times 10^{-01}$ | $2.81 \times 10^{-01}$ | |
| **Clinical examination** | | | | | | | | |
| Oxygen saturation | 95.0 (90.0, 97.0) [783] | 0.37 (0.30, 0.45) | $6.42 \times 10^{-20}$ | 97.0 (94.0, 99.0) [132] | 0.35 (0.20, 0.63) | $4.33 \times 10^{-04}$ | $7.32 \times 10^{-22}$ | Yes |
| Diastolic pressure | 80.0 (69.0, 90.0) [769] | 0.70 (0.60, 0.83) | $3.93 \times 10^{-05}$ | 78.0 (69.0, 84.8) [138] | 0.75 (0.51, 1.11) | $1.55 \times 10^{-01}$ | $1.74 \times 10^{-05}$ | Yes |
| Respiratory rate | 25.0 (20.0, 30.0) [667] | 1.33 (1.12, 1.58) | $1.10 \times 10^{-03}$ | 22.0 (18.0, 28.0) [67] | 3.37 (1.28, 8.86) | $1.39 \times 10^{-02}$ | $2.70 \times 10^{-04}$ | Yes |
| Systolic pressure | 134.0 (118.0, 148.0) [769] | 0.82 (0.70, 0.97) | $1.84 \times 10^{-02}$ | 125.0 (113.5, 137.5) [139] | 0.74 (0.50, 1.10) | $1.37 \times 10^{-01}$ | $9.87 \times 10^{-03}$ | |
| Cardiac frequency | 95.0 (82.0, 107.0) [771] | 0.94 (0.80, 1.10) | $4.29 \times 10^{-01}$ | 90.0 (80.0, 102.2) [136] | 1.27 (0.86, 1.87) | $2.33 \times 10^{-01}$ | $5.68 \times 10^{-01}$ | |
| Body temperature | 37.8 (37.0, 38.5) [788] | 0.99 (0.84, 1.17) | $9.28 \times 10^{-01}$ | 37.3 (36.7, 38.0) [137] | 6.09 (0.97, 38.36) | $5.44 \times 10^{-02}$ | $8.07 \times 10^{-01}$ | |
| **Symptoms** | | | | | | | | |
| Dyspnea | 68% [837] | 1.73 (1.23, 2.45) | $1.87 \times 10^{-03}$ | 57% [150] | 1.67 (0.78, 3.59) | $1.88 \times 10^{-01}$ | $9.93 \times 10^{-04}$ | |
| Chest pain | 8.2% [837] | 0.27 (0.12, 0.61) | $1.54 \times 10^{-03}$ | 5.3% [150] | 0.78 (0.15, 4.04) | $7.68 \times 10^{-01}$ | $1.52 \times 10^{-03}$ | |
| Myalgia | 24% [837] | 0.59 (0.41, 0.86) | $6.20 \times 10^{-03}$ | 13% [150] | 0.99 (0.32, 3.01) | $9.83 \times 10^{-01}$ | $6.95 \times 10^{-03}$ | |
| Confusion | 9.4% [837] | 1.73 (1.06, 2.81) | $2.80 \times 10^{-02}$ | 2.7% [150] | 7.62 (0.77, 75.63) | $8.28 \times 10^{-02}$ | $1.37 \times 10^{-02}$ | |
| Coughing | 41% [837] | 0.78 (0.57, 1.06) | $1.10 \times 10^{-01}$ | 31% [150] | 0.88 (0.40, 1.97) | $7.64 \times 10^{-01}$ | $1.04 \times 10^{-01}$ | |
| Diarrhea | 8.0% [837] | 1.30 (0.76, 2.24) | $3.42 \times 10^{-01}$ | 6.0% [150] | 1.20 (0.21, 6.82) | $8.39 \times 10^{-01}$ | $3.31 \times 10^{-01}$ | |
| Fever | 51% [837] | 1.11 (0.82, 1.50) | $4.98 \times 10^{-01}$ | 37% [150] | 1.53 (0.71, 3.26) | $2.75 \times 10^{-01}$ | $3.92 \times 10^{-01}$ | |
| Asthenia | 17% [837] | 0.78 (0.51, 1.18) | $2.41 \times 10^{-01}$ | 8.0% [150] | 3.18 (0.91, 11.09) | $7.01 \times 10^{-02}$ | $4.01 \times 10^{-01}$ | |
| Symptoms duration before examination | 6.0 (3.0, 9.0) [771] | 1.02 (0.87, 1.19) | $8.13 \times 10^{-01}$ | 4.0 (2.0, 7.8) [126] | 1.15 (0.78, 1.68) | $4.76 \times 10^{-01}$ | $7.21 \times 10^{-01}$ | |
| Dry quintuous cough | 33% [837] | 0.95 (0.69, 1.31) | $7.43 \times 10^{-01}$ | 20% [150] | 1.23 (0.48, 3.17) | $6.62 \times 10^{-01}$ | $8.05 \times 10^{-01}$ | |
| Headache | 5.3% [837] | 1.09 (0.57, 2.09) | $8.04 \times 10^{-01}$ | 3.3% [150] | 0.58 (0.06, 5.39) | $6.35 \times 10^{-01}$ | $8.72 \times 10^{-01}$ | |
| **Comorbidities and smoking** | | | | | | | | |
| Chronic kidney disease | 12% [837] | 2.39 (1.54, 3.70) | $9.51 \times 10^{-05}$ | 7.3% [150] | 16.59 (1.93, 142.84) | $1.06 \times 10^{-02}$ | $1.81 \times 10^{-05}$ | Yes |
| Hypertension | 45% [837] | 1.77 (1.31, 2.40) | $2.43 \times 10^{-04}$ | 35% [150] | 1.11 (0.51, 2.42) | $7.9 \times 10^{-01}$ | $2.52 \times 10^{-04}$ | Yes |
| Asthma | 8.7% [837] | 0.41 (0.21, 0.80) | $8.87 \times 10^{-03}$ | 6.0% [150] | 0.32 (0.04, 2.71) | $2.97 \times 10^{-01}$ | $5.81 \times 10^{-03}$ | Yes |
| Cardiac disease | 22% [837] | 1.41 (0.99, 2.02) | $5.77 \times 10^{-02}$ | 17% [150] | 1.05 (0.39, 2.78) | $9.26 \times 10^{-01}$ | $5.94 \times 10^{-02}$ | |
| Diabetes | 24% [837] | 1.32 (0.94, 1.87) | $1.13 \times 10^{-01}$ | 19% [150] | 2.10 (0.83, 5.30) | $1.15 \times 10^{-01}$ | $6.66 \times 10^{-02}$ | |
| Smoker | 15% [837] | 1.32 (0.87, 2.00) | $1.88 \times 10^{-01}$ | 25% [150] | 2.85 (1.26, 6.44) | $1.17 \times 10^{-02}$ | $8.30 \times 10^{-02}$ | |
| Emphysema | 6.0% [837] | 1.47 (0.80, 2.71) | $2.12 \times 10^{-01}$ | 14% [150] | 4.12 (1.51, 11.25) | $5.62 \times 10^{-03}$ | $8.73 \times 10^{-02}$ | |
| Corticosteroids | 3.9% [837] | 0.61 (0.26, 1.43) | $2.57 \times 10^{-01}$ | 12% [150] | 3.14 (1.05, 9.37) | $3.99 \times 10^{-02}$ | $4.48 \times 10^{-01}$ | |
| Chemotherapy | 1.0% [837] | 1.39 (0.33, 5.88) | $6.50 \times 10^{-01}$ | 39% [150] | 1.88 (0.89, 3.98) | $9.94 \times 10^{-02}$ | $4.64 \times 10^{-01}$ | |
| Dyslipidemia | 16% [837] | 1.09 (0.73, 1.65) | $6.71 \times 10^{-01}$ | 11% [150] | 2.67 (0.87, 8.18) | $8.65 \times 10^{-02}$ | $4.74 \times 10^{-01}$ | |
| NSAI | 3.6% [837] | 1.23 (0.56, 2.68) | $6.05 \times 10^{-01}$ | 0.7% [150] | 0 (0, Inf) | $9.88 \times 10^{-01}$ | $6.13 \times 10^{-01}$ | |
| Cancer | 7.3% [837] | 1.17 (0.66, 2.08) | $5.87 \times 10^{-01}$ | 85% [150] | 0.47 (0.17, 1.30) | $1.45 \times 10^{-01}$ | $7.78 \times 10^{-01}$ | |
| **Biological measures** | | | | | | | | |
| LDH | 341.0 (263.5, 452.5) [527] | 2.02 (1.63, 2.51) | $1.51 \times 10^{-10}$ | 278.0 (203.8, 400.0) [134] | 2.36 (1.32, 4.21) | $3.77 \times 10^{-03}$ | $9.78 \times 10^{-12}$ | Yes |
| Urea | 6.0 (4.2, 9.8) [739] | 1.66 (1.40, 1.97) | $5.72 \times 10^{-09}$ | 5.1 (3.7, 6.9) [117] | 2.19 (1.36, 3.52) | $1.28 \times 10^{-03}$ | $3.07 \times 10^{-10}$ | Yes |
| CRP | 69.0 (29.0, 130.0) [729] | 1.48 (1.27, 1.74) | $1.07 \times 10^{-06}$ | 52.7 (15.1, 115.7) [142] | 1.48 (1.03, 2.14) | $3.64 \times 10^{-02}$ | $2.37 \times 10^{-07}$ | Yes |
| Neutrophil | 5.0 (3.5, 7.1) [746] | 1.38 (1.18, 1.62) | $8.31 \times 10^{-05}$ | 4.4 (2.3, 7.1) [148] | 1.15 (0.80, 1.64) | $4.51 \times 10^{-01}$ | $6.16 \times 10^{-05}$ | Yes |

## Table 1 (continued)

| | KB, N = 837 | | | IGR, N = 150 | | | Pooled | |
|---|---|---|---|---|---|---|---|---|
| | Distribution | Odds ratio | p-value association with severity | Distribution | Odds ratio | p-value association with severity | Pooled p-value | Significant association |
| Leukocytes | 6.7 (4.9, 9.0) [750] | 1.31 (1.12, 1.54) | $7.18 \times 10^{-04}$ | 6.6 (4.1, 9.8) [144] | 1.19 (0.83, 1.71) | $3.43 \times 10^{-01}$ | $4.73 \times 10^{-04}$ | Yes |
| Platelet | 206.0 (161.5, 259.5) [751] | 0.78 (0.66, 0.93) | $5.47 \times 10^{-03}$ | 213.5 (156.0, 308.0) [144] | 0.76 (0.50, 1.15) | $1.93 \times 10^{-01}$ | $3.06 \times 10^{-03}$ | |
| Ferritin | 627.0 (287.8, 1285.5) [296] | 1.27 (0.99, 1.62) | $6.33 \times 10^{-02}$ | 567.0 (233.0, 1327.0) [125] | 1.52 (1.03, 2.26) | $3.70 \times 10^{-02}$ | $2.85 \times 10^{-02}$ | |
| Total bilirubin | 8.0 (6.0, 11.0) [691] | 1.16 (0.99, 1.37) | $6.80 \times 10^{-02}$ | 9.0 (6.0, 13.0) [137] | 1.52 (1.04, 2.23) | $3.25 \times 10^{-02}$ | $3.01 \times 10^{-02}$ | |
| Conjugated bilirubin | 12.0 (9.0, 16.0) [57] | 1.32 (0.72, 2.42) | $3.63 \times 10^{-01}$ | 2.0 (2.0, 5.0) [68] | 2.00 (1.10, 3.65) | $2.29 \times 10^{-02}$ | $1.97 \times 10^{-01}$ | |
| Lymphocyte | 1.0 (0.7, 1.4) [745] | 0.89 (0.70, 1.12) | $3.16 \times 10^{-01}$ | 1.0 (0.6, 1.6) [148] | 0.80 (0.53, 1.21) | $2.89 \times 10^{-01}$ | $2.42 \times 10^{-01}$ | |
| Monocyte | 0.5 (0.3, 0.6) [746] | 0.93 (0.79, 1.10) | $3.99 \times 10^{-01}$ | 0.5 (0.3, 0.8) [148] | 0.66 (0.36, 1.19) | $1.64 \times 10^{-01}$ | $2.84 \times 10^{-01}$ | |
| Hemoglobin | 13.2 (12.1, 14.4) [751] | 1.06 (0.90, 1.24) | $4.84 \times 10^{-01}$ | 11.1 (9.5, 12.8) [144] | 0.65 (0.43, 0.98) | $3.79 \times 10^{-02}$ | $7.43 \times 10^{-01}$ | |
| Creatine kinase | 146.0 (84.0, 312.8) [638] | 0.98 (0.82, 1.18) | $8.47 \times 10^{-01}$ | 70.5 (30.8, 157.2) [92] | 1.32 (0.83, 2.10) | $2.36 \times 10^{-01}$ | $9.88 \times 10^{-01}$ | |
| Radiological standardized report | | | | | | | | |
| Disease extent 0/1/2/3/4/5 | 6.3/17/37/27/11/2.6% [806] | 2.43 (2.01, 2.93) | $2.13 \times 10^{-20}$ | 14/36/24/7.2/10/8.7% [138] | 1.62 (1.11, 2.37) | $1.21 \times 10^{-02}$ | $1.28 \times 10^{-21}$ | Yes |
| Crazy paving | 44% [799] | 2.72 (1.98, 3.74) | $6.48 \times 10^{-10}$ | 40% [140] | 2.37 (1.10, 5.11) | $2.72 \times 10^{-02}$ | $9.94 \times 10^{-11}$ | Yes |
| Peripheral distribution | 63% [749] | 0.52 (0.38, 0.72) | $7.84 \times 10^{-05}$ | 32% [137] | 0.55 (0.23, 1.31) | $1.77 \times 10^{-01}$ | $3.73 \times 10^{-05}$ | Yes |
| Predominance inferior | 40% [732] | 1.46 (1.06, 2.02) | $2.13 \times 10^{-02}$ | 43% [136] | 0.90 (0.42, 1.93) | $7.81 \times 10^{-01}$ | $2.65 \times 10^{-02}$ | |
| GGO | 91% [824] | 1.72 (0.94, 3.17) | $8.02 \times 10^{-02}$ | 76% [140] | 1.55 (0.60, 3.98) | $3.66 \times 10^{-01}$ | $6.01 \times 10^{-02}$ | |
| Consolidation | 69% [773] | 1.36 (0.95, 1.94) | $9.23 \times 10^{-02}$ | 39% [140] | 1.13 (0.53, 2.41) | $7.56 \times 10^{-01}$ | $8.69 \times 10^{-02}$ | |
| GGO rounded | 15% [750] | 0.64 (0.40, 1.03) | $6.63 \times 10^{-02}$ | 7.5% [107] | 3.10 (0.65, 14.75) | $1.56 \times 10^{-01}$ | $1.18 \times 10^{-01}$ | |
| Consolidation rounded | 17& [535] | 1.08 (0.66, 1.77) | $7.59 \times 10^{-01}$ | 18% [55] | 0.92 (0.21, 4.09) | $9.11 \times 10^{-01}$ | $7.77 \times 10^{-01}$ | |
| Other radiological patterns | | | | | | | | |
| Cardiomegaly | 25% [115] | 2.84 (1.19, 6.80) | $1.89 \times 10^{-02}$ | 23% [78] | 0.77 (0.22, 2.72) | $7.40 \times 10^{-01}$ | $2.42 \times 10^{-02}$ | |
| Splenomegaly | 2.6% [115] | 4.11 (0.36, 46.85) | $2.55 \times 10^{-01}$ | 10% [78] | 17.33 (1.88, 160) | $1.40 \times 10^{-02}$ | $1.22 \times 10^{-01}$ | |
| Hepatomegaly | 8.7% [115] | 1.35 (0.36, 5.12) | $6.56 \times 10^{-01}$ | 18% [78] | 5.17 (1.41, 18.99) | $2.02 \times 10^{-02}$ | $4.00 \times 10^{-01}$ | |
| COVID treatment | | | | | | | | |
| Corticoid | 1.0% [837] | | | 4.7% [150] | | | | |
| Hydroxychloroquine | 4.1% [837] | | | 34% [150] | | | | |
| Immunomodulator | 2.5% [837] | | | 0% [144] | | | | |
| Kaleta | 1.4% [837] | | | 2.0% [150] | | | | |
| Anti-IL6 | 3.2% [837] | | | 4.7% [150] | | | | |
| Interferon | 0.6% [837] | | | 0% [144] | | | | |

Among the 1003 patients of the study, biological and clinical variables were available for 987 individuals. Categorical variables are expressed as percentages [available]. Continuous variables are shown as median (IQR) [available]. Associations with severity are evaluated with logistic regression and reported with two-sided p-values for each center and p-values were combined with Stouffer's method. The column entitled "Significant association" indicates that the variable is significantly associated with severity after Bonferroni adjustment to account for multiple testing across 58 variables (treatments are excluded). For continuous variables, odds ratios are computed for an increase of one standard deviation of the continuous variable.

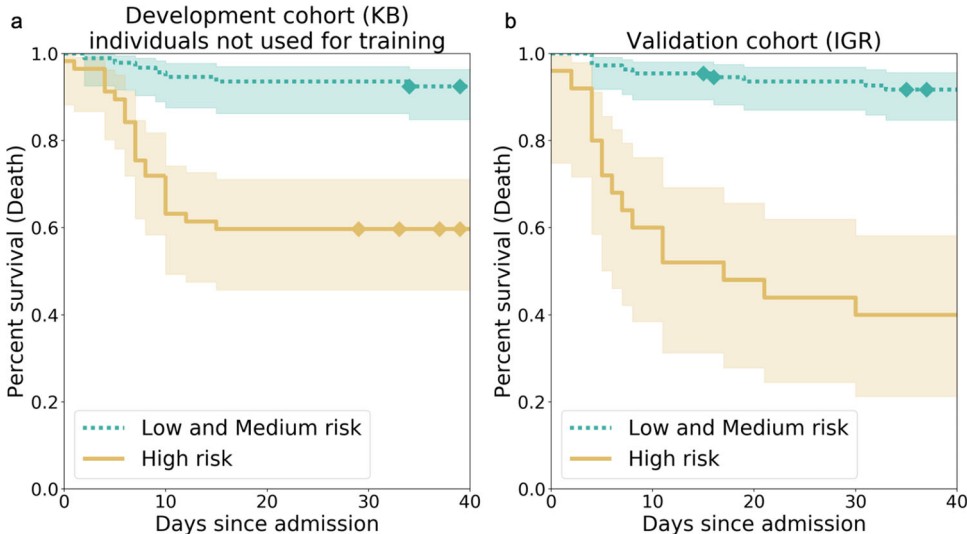

**Fig. 1 Kaplan–Meier curves for the high-risk individuals and the ones with low or medium risk according to AI-severity.** The threshold to assign individuals into a high-risk group was the 2/3 quantile of the AI-severity score computed for patients of the KB development cohort. **a** Kaplan–Meier curves were obtained for the 150 leftover KB patients from the development cohort. **b** Kaplan–Meier curves were obtained for the 135 patients of the IGR validation cohort. p-values for the log-rank test were equal to 4.77e−07 (KB) and 4.00e−12 (IGR). The two terciles used to determine threshold values for low-, medium-, and high-risk groups were equal to 0.187 and 0.375. Diamonds correspond to censoring of patients who were still hospitalized at the time when data ceased to be updated. The bands correspond to the sequence of the 95% confidence intervals of the survival probabilities for each day. KB Kremlin-Bicêtre hospital, IGR Institut Gustave Roussy hospital.

**Table 2 Statistical measures of the performance of AI-severity.**

|  | AUC | Sensitivity | Specificity | PPV | NPV |
|---|---|---|---|---|---|
| | $O_2 \geq 15$ L/min or Ventilation or Death | | | | |
| Development cohort (KB) | 0.77 (0.69–0.86) | 0.70 (0.56–0.83) | 0.75 (0.66–0.84) | 0.54 (0.40–0.67) | 0.86 (0.78–0.93) |
| Validation cohort (IGR) | 0.79 (0.70–0.87) | 0.47 (0.30–0.63) | 0.94 (0.89–0.98) | 0.76 (0.56–0.92) | 0.81 (0.73–0.88) |
| | Death or ICU | | | | |
| Development cohort (KB) | 0.79 (0.71–0.86) | 0.68 (0.54–0.80) | 0.77 (0.68–0.85) | 0.60 (0.46–0.73) | 0.83 (0.74–0.90) |
| Validation cohort (IGR) | 0.86 (0.78–0.92) | 0.50 (0-.34–0.65) | 0.96 (0.91–0.99) | 0.84 (0.68–0.96) | 0.81 (0.72–0.88) |
| | Death | | | | |
| Development cohort (KB) | 0.81 (0.73–0.89) | 0.77 (0.60–0.91) | 0.72 (0.62–0.80) | 0.40 (0.27–0.53) | 0.92 (0.86–0.98) |
| Validation cohort (IGR) | 0.88 (0.81–0.94) | 0.62 (0.42–0.80) | 0.91 (0.84–0.96) | 0.60 (0.39–0.79) | 0.92 (0.86–0.96) |

To compute sensitivity, specificity, PPV, and NPV, we assigned individuals into a high-risk group or a low- and medium-risk group depending on their AI-severity score. The threshold to assign individuals into a high-risk group was the 2/3 quantile of the AI-severity scores computed for patients of the KB training set. To compute measures of performance for the KB development cohort, we considered the 150 leftover individuals who were not used to learn the coefficients of AI-severity. KB Kremlin-Bicêtre hospital, IGR Institut Gustave Roussy hospital, PPV positive predictive value, NPV negative predictive value, ICU intensive care unit.

severity outcome of "oxygen flow rate of 15 L/min or higher, or need for mechanical ventilation, or death." All the prognosis scores were also evaluated on two other outcomes that consist of "death or ICU admission" and "death."

We evaluated AI-severity with several statistical measures of performance. The discriminative ability of AI-severity was of AUC = 0.78 (0.69, 0.86) for the 150 leftover KB patients, and of AUC = 0.79 (0.70, 0.87) for the validation IGR dataset. We also evaluated calibration properties of AI-severity using calibration plot (Supplementary Fig. 3)[24]. We found slope of 0.949 (0.650, 1.371) (150 leftover individuals at KB) and of 0.996 (0.755, 1.383) (IGR), and intercept (calibration-in-the-large) of −0.206 (−0.564, 0.172) (KB) and of 0.529 (0.088, 1.084) (IGR). Estimated slopes and intercepts indicated correct calibration of AI-severity for the leftover patients of the development KB cohort and an under-estimation of severe outcomes for the validation IGR cohort; AI-severity predicted a mean severity of 22% (0.18, 0.25) for the 135 IGR patients, whereas severe outcomes occurred for 30% (0.22, 0.37) of these patients.

To compute additional measures of performance, individuals in the top tercile were assigned in a high-risk group. We found that the survival function of the individuals at high risk was significantly different from the survival function of the other individuals (Fig. 1, p = 4.77e−07 at KB, p = 4.00e−12 at IGR for a log-rank test). When considering a binary classification consisting of a high-risk group and a medium- or low-risk group, we obtained for the "$O_2 \geq 15$ L/min or Ventilation or Death" outcome, a positive predictive values (or precision) of 54% (0.40–0.67) (KB) and 76% (0.56–0.92) (IGR), negative predictive values of 86% (0.78–0.93) (KB) and 81% (0.73–0.88) (IGR), specificities of 75% (0.66–0.84) (KB) and 94% (0.89–0.98) (IGR), and sensitivities of 70% (0.56–0.83) (KB) and 47% (0.30–0.63) (IGR) (Table 2).

AI-severity outperformed 11 previously published severity or mortality scores that were developed using 200–50,000 patients in the development and validation cohorts (Fig. 2 and Supplementary Table 3). The mean difference (averaged over outcomes) between the AUC of AI-severity and of other scores ranged

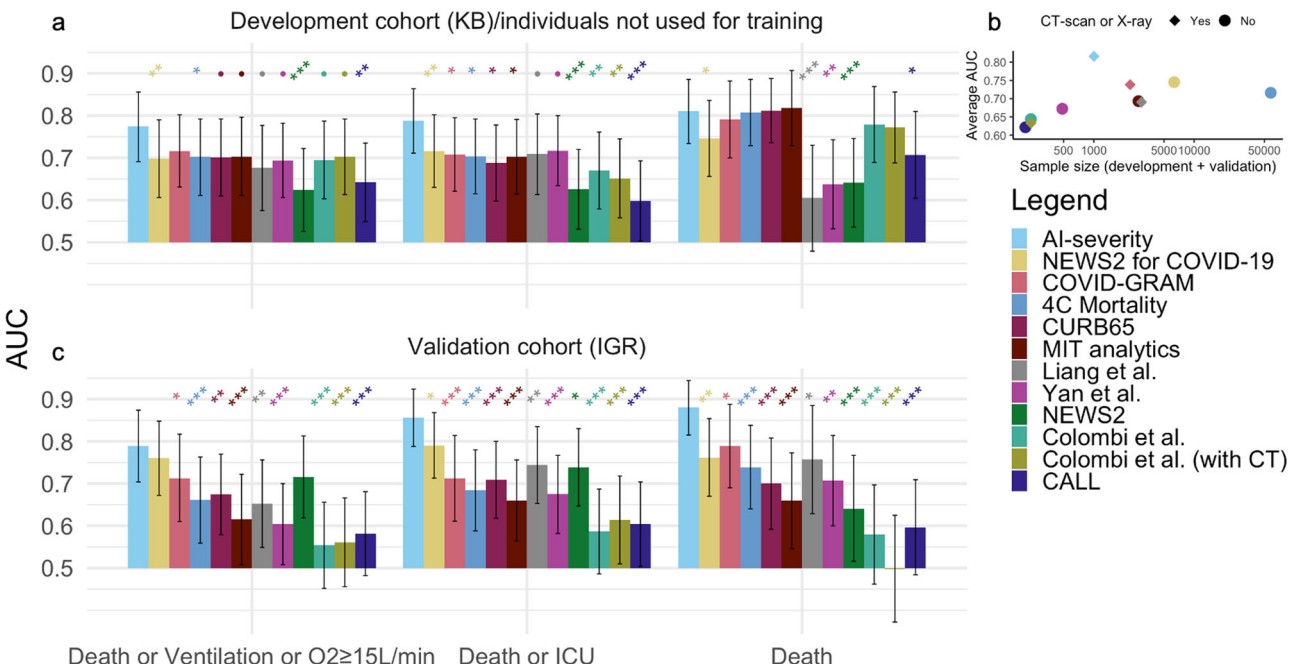

**Fig. 2 AUC values when comparing AI-severity to other prognostic scores for COVID-19 severity/mortality.** The AI-severity model was trained using the severity outcome defined as an oxygen flow rate of 15 L/min or higher, the need for mechanical ventilation, or death. When evaluating AI-severity on the alternative outcomes, the model was not trained again. **a** AUC results are reported on the leftover KB patients from the development cohort (150 patients). **b** The mean AUC (averaged over outcomes and over hospitals) as a function of the sample size (sum of sample sizes for the development and validation cohorts) used to construct the score. **c** AUC results are reported on the external validation set from IGR (135 patients). Models are sorted from left to right (and from top to bottom in the legend) by decreasing order of AUC values (averaged over outcomes and over hospitals). Error bars represent the 95% confidence intervals obtained with the DeLong procedure. Stars indicate the order of magnitude of *p*-values for the DeLong one-sided test in which we test if $\text{AUC}_{\text{AI-severity}} > \text{AUC}_{\text{other score}}$, ● $0.05 < p \leq 0.10$, *$0.01 < p \leq 0.05$, **$0.001 < p \leq 0.01$, ***$p \leq 0.001$. KB Kremlin-Bicêtre hospital, IGR Institut Gustave Roussy hospital, ICU intensive care unit, NEWS2 National Early Warning Score 2, AUC area under the curve.

between 0.05 (4C mortality, COVID-GRAM, CURB-65, MIT analytics) and 0.16 (NEWS2) for the 150 leftover patients of the KB development cohort and between 0.07 (NEWS2 for COVID-19) and 0.28[16] for the 135 patients of the IGR validation cohort. Among alternative scores, the COVID-GRAM, the NEWS2 for COVID-19 score, and the 4C mortality scores were the ones with the largest mean AUC values (averaged over outcomes and hospitals). The AI-severity score was significantly larger than the NEWS2 for COVID-19 score for all outcomes when evaluated with the leftover patients of the KB development cohort and for the "Death or ICU" and the "Death" outcomes when evaluated with patients from the IGR validation cohort. Differences between AI-severity on the one hand and the COVID-GRAM score or the 4C mortality score on the other hand were significant only for the "Death or ICU" outcome when being evaluated on the leftover patients of the KB development cohort but they were significant for all outcomes when being evaluated on the validation IGR cohort.

**Development of alternative models that include CT scan information.** In addition to AI-severity, we considered two alternative scores that also integrate CT scan information. The two scores include the same clinical and biological variables (age, sex, oxygen saturation, urea, platelets) as AI-severity. The first score (AI-segment) uses an automatic quantification of disease extent to include CT scan information and the second score (C & B & RR) considers a radiologist quantification—available in the radiological report—instead. AI-segment relies on segmentation of lesions that was performed by training another deep learning model using fully annotated and partially annotated CT scans

(Supplementary Notes). The correlation between automatic quantification of lung lesions with AI-segment and radiologist quantification was of 0.56 (Supplementary Fig. 4 and Supplementary Notes).

AI-severity has a superior discriminative ability when compared to the alternative C & B & RR and AI-segment scores, although differences of AUC were generally not significant (Supplementary Fig. 5). The mean difference averaged over outcomes between $\text{AUC}_{\text{AI-severity}}$ and $\text{AUC}_{\text{C \& B \& RR}}$ (resp. $\text{AUC}_{\text{AI-segment}}$) is null (resp. 0.03) for the 150 leftover KB patients of the development cohort and of 0.04 (resp. 0.01) for the IGR validation cohort. Differences between scores were not significant except when comparing AI-severity to AI-segment at KB for the "Death or ICU" outcome (Supplementary Fig. 5).

**Additional value of CT scan information.** Last, we evaluate to what extent CT scan adds prognosis information to the clinical characteristics and biological variables from lab tests. To this end, we trained a score named C & B based on clinical and biological variables only. The AUC of the scores that integrate CT scan information was larger or equivalent to the AUC of the C & B score (Supplementary Fig. 6). The mean difference averaged over outcomes between $\text{AUC}_{\text{AI-severity}}$ and $\text{AUC}_{\text{C\&B}}$ was equal to 0.03 for both cohorts. Differences between AI-severity and C & B were significant for some outcomes and cohorts but not for all combinations (Supplementary Fig. 6). We also computed the confusion matrix for the outcome "oxygen flow rate of 15 L/min or higher and/or the need for mechanical ventilation and/or patient death" (Fig. 3). AI-severity correctly classified 3 and 4 additional positive patients among the 44 and 40 positive patients of the

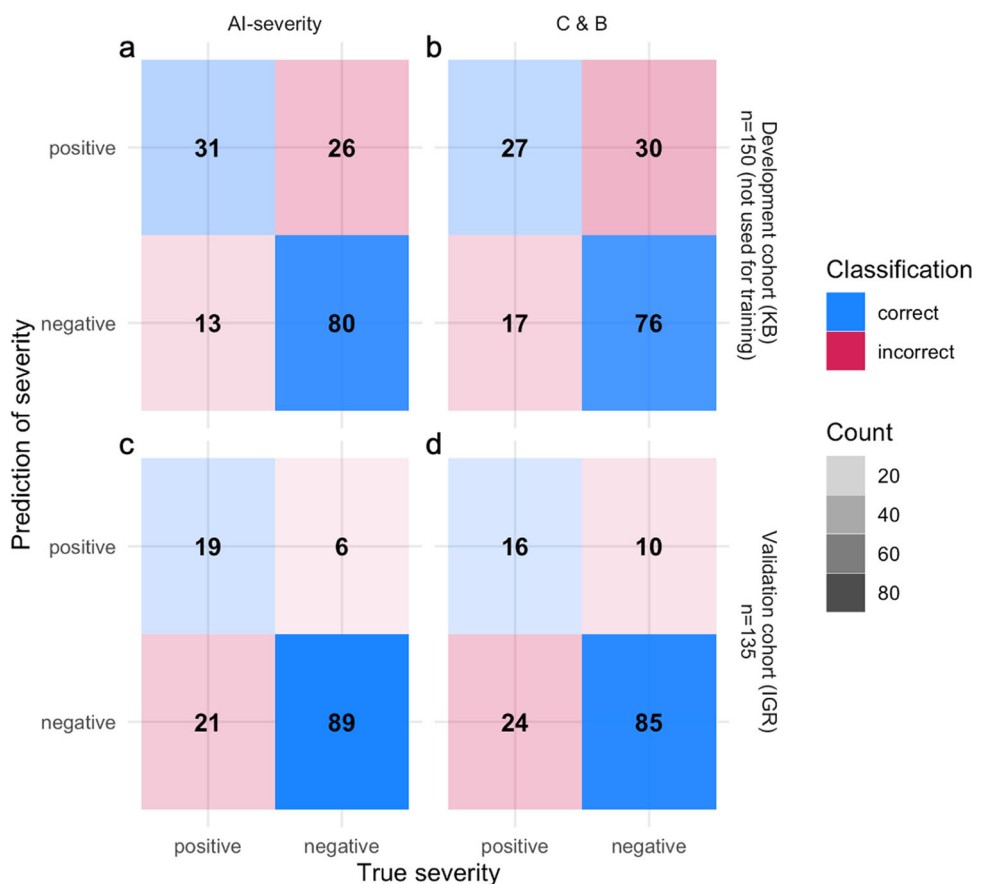

**Fig. 3 Confusion matrix obtained with AI-severity, which includes CT scan information in addition to clinical and biological variables and with C & B, which contains only clinical and biological variables.** Values in the matrices correspond to the number of patients in each category, which is defined by the true severity status and its predicted one. The confusion matrix was computed using the outcome "oxygen flow rate of 15 L/min or higher and/or the need for mechanical ventilation and/or patient death." For both scores, we considered the 2/3 quantile—computed using the development cohort (KB)—to distinguish severe patients from non-severe patients. In addition to the neural network variable computed from CT scan images, the variables included in AI-severity consist of oxygen saturation, age, sex, platelet, and urea. The variables included in C & B consist of oxygen saturation, age, sex, platelet, urea, LDH, hypertension, chronic kidney disease, dyspnea, and neutophil values. Both scores were constructed using a feature selection algorithm that selected optimal variables. KB Kremlin-Bicêtre hospital, IGR Institut Gustave Roussy hospital.

development and validation cohorts when compared to C & B and 4 additional negative patients among the 106 and 95 negative patients of the cohorts. Overall, CT scan information increases AUC by a measurable but limited amount in both cohorts; there was a difference of AUC of 0.03 when comparing AI-severity to the C & B score.

To interpret the difference of AUC, we computed differences of AUC for several subgroups of patients. Because CT scan information is correlated with markers of inflammation[25], we considered subgroups of patients with different levels of inflammation. The difference of AUC was significantly larger in patients with higher levels of inflammatory markers for 150 leftover patients of the development cohort (KB) (paired $t$-test, $p = 0.003$) but the difference was not significant for the validation cohort (IGR) (paired $t$-test, $p = 0.24$) (Supplementary Fig. 7). In both cohorts, the subgroup analysis suggested that prognosis of patients with larger values of CRP, LDH, and leukocytes benefited from the inclusion of CT scan information (Supplementary Fig. 7).

To further investigate the added prognosis value of CT scan, we studied the association between COVID-19 severity and the prognosis variable provided by the neural network. In the KB dataset, the three variables that were the most correlated with the prognosis variable of the neural network were oxygen saturation ($r = -0.52$ ($-0.58, -0.48$)), LDH ($r = 0.46$ ($0.39, 0.52$)), and CRP

($r = 0.43$ ($0.37, 0.49$)) (Supplementary Table 4). To account for the confounding effect of these variables, we regressed the severity outcome with the neural network prognosis variable and the three correlated variables. We found that the neural network variable was significantly correlated with the severity outcome ($p = 0.01$). The statistical evidence for association between the neural network prognosis variable and COVID-19 severity was also found ($p = 3.24 \times 10^{-6}$) when accounting for the five additional variables of AI-severity. This confirms that CT scan information captured by the neural network brings unique prognostic information.

## Discussion

Using a deep learning model to capture CT scan prognosis information, we have built the AI-severity score to prognose severe evolution for COVID-19 hospitalized patients. In addition to the deep learning variable, AI-severity is based on age, sex, oxygen saturation, urea, and platelet counts. On the IGR validation cohort containing a majority of cancer patients, AI-severity provided values of AUCs significantly larger than the ones obtained with the best prognosis scores of our comparative analysis, which consist of COVID-GRAM, the NEWS2 score modified for COVID-19 patients, and the 4C mortality score[6,12,26]. Taken together, these results show that future disease

severity markers are present within routine CT scans performed at admission.

Looking back on the prognostic clinical and biological variables, we found 12 of these significantly associated with severe evolution, which is consistent with previous studies[15,27,28]. First, looking at clinical characteristics, we confirmed that male and older persons are more at risk[29]. Second, looking at clinical examination variables, we found that respiratory rate, diastolic pressure, and oxygen saturation are clinical variables associated with severity. These associations may reflect physician decisions taken for ICU triage. Inclusion criteria for critical care triage include (i) requirement for invasive ventilatory support characterized by an oxygen saturation lower than 90%, or by respiratory failure, or (ii) requirement for vasopressors characterized by hypotension and low blood pressure[30]. Third, looking at comorbidities, we confirmed the results of several meta-analyses[28,31–33] that showed that chronic kidney disease and hypertension are linked to severity. We however did not find significant associations for other comorbidities previously associated with severity, such as diabetes, and cardiovascular diseases[33,34]. While we expected cancer patients to have more severe outcomes because they are generally older, with multiple comorbidities and often in a treatment-induced immunosuppressive state[35–37], we did not find this association. Several factors can explain this. Each cohort was not optimally balanced to conclusively study the association between cancer and severity: IGR admitted mostly cancer patients (80% of the patients), while KB admitted very few cancer patients (7%). Fourth, looking at COVID-19 symptoms, we did not find any significantly associated with severity. Dyspnea is a prominent symptom that has been repeatedly associated with severity and our results are compatible with a positive association with severity but we may lack a large-enough sample size to be significant[6,38,39]. Last, looking at biological measures, we found that inflammatory biomarkers, LDH, and CRP are related to severity[14,27,40]. We also found association of severity with leukocytes, neutrophils, and urea, the latter being explained by the fact that high urea is indicative of kidney dysfunction. Thrombocytopenia (low platelet count) was not significantly associated with severity, possibly because of lack of statistical power and stringent correction for multiple testing, but association between thrombocytopenia and severity was in the expected direction and platelet counts are included in the 6-variable AI-severity score[41].

Beyond these clinical and biological variables, chest CT scans provided additional markers of disease severity. Significant features include the total extent of lesions, and the presence of crazy-paving pattern lesions. Although the extent of disease severity and consolidation are known to be associated with severity[16,19,42–47], our study discovered its association with crazy paving, a precursor of consolidation lesions. Initial damages to the alveoli, as well as protein and fibrous exudation, explain the early onset of GGO. As the disease progresses, more and more inflammatory cells infiltrate the alveoli and interstitial space, followed by diffuse alveolar lesions and the formation of a hyaline membrane, which results in a crazy-paving appearance, which is then followed by consolidation on the CT examination[48,49].

Compared to a radiologist's reporting and quantification of lesions, there are several advantages to capturing CT scan information through a deep learning model. Good reproducibility is a key element for imaging biomarkers, and visual inspection of images introduces variability that can hinder its clinical application[50]. Another advantage is that radiologists are faced with the challenge that large numbers of cases must be read, annotated, and prioritized in a COVID-19 pandemic. AI analysis of radiological images has the potential to reduce this burden and speed up their reading time. Finally, prognosis scores obtained with

deep learning models trained on CT scans are more predictive of severity than a quantification of disease extent performed by a radiologist. We indeed showed that internal representation of the AI-severity neural network model captures clinical information from CT scans, and this can be particularly useful when some clinical or lab measurements are missing.

Our reported prognostic values for CT scan-based models (AUC range of 0.70–0.80) are lower than the 0.85 AUC reported in a previously published study that uses deep learning with CT scan images for prognosis[17]. We hypothesize that this is due to use of different outcome definitions, as well as different patient characteristics in the study cohorts (age, severity at admission, etc.). Hospital admission criteria vary between countries and hospitals; for instance, the proportion of deaths in our French KB and IGR cohorts was of 16–17%, while it was of 39% in the study that reported larger AUC values[17]. When applying other previously published scores to the KB and IGR datasets, we found smaller AUC scores than reported values in the original papers. This difference can again be explained by differing patient characteristics, and different measures of severity between studies[6,7,9,10,16].

Our evaluation of AI-severity and of alternative scores revealed that including CT scan information in addition to clinical and biological information significantly improves prognosis of future severity at least for the IGR validation cohort. A better prognosis performance was more pronounced for subgroups containing patients with higher levels of inflammatory markers. The neural network prognosis variable was correlated with biological and clinical severity biomarkers such as CRP levels, tissue damage (LDH), and oxygenation. Information redundancy between data modalities explains the relatively modest 0.03 increase of AUC values provided by CT scan when being added to biological and clinical variables[25,38,51–53].

Beyond AI modeling, our study shows that the 6-variable AI-severity score integrating a radiological quantification of lesions with key clinical and biological variables provides accurate severity predictions. When comparing AI-severity with 11 existing scores for severity, we find significantly improved prognosis performance in the validation datasets of 150 and 135 patients. Our results suggest that AI-severity can become a useful severity scoring approach for COVID-19 patients.

## Methods

**Description of the retrospective study**. Data including CT scans were collected at two French hospitals (Kremlin-Bicêtre Hospital, APHP, Paris denoted as KB and Gustave Roussy Hospital, Villejuif denoted as IGR). CT scans, clinical, and biological data were collected in the first 2 days after hospital admission. This study has received approval of ethic committees from the two hospitals and authors submitted a declaration to the National Commission of Data Processing and Liberties (N° INDS MR5413020420, CNIL) in order to get registered in the medical studies database and respect the General Regulation on Data Protection (RGPD) requirements. An information letter was sent to all patients included in the study. We stopped to update information about patient status on 5 May. Among the 1003 patients of the study, two patients asked to be excluded from the study.

Inclusion criteria were (1) date of admission at hospital (from 2 February to 20 March at Kremlin-Bicêtre and from the 2 March to 24 April at Institut Gustave Roussy) and (2) a positive diagnosis of COVID-19. Patients were considered positive either because of a positive real-time fluorescence polymerase chain reaction (RT-PCR) based on nasal or lower respiratory tract specimens or a CT scan with a typical appearance of COVID-19 as defined by the ACR criteria for negative RT-PCR patients[54]. Children and pregnant women were excluded from the study.

The clinical and laboratory data were obtained from detailed medical records, cleaned and formatted retrospectively by ten radiologists with 3–20 years of experience (five radiologists at GR and five at KB). Data include demographic variables: age and sex, variables from the clinical examination include: body weight and height, body mass index, heart rate, body temperature, oxygen saturation, blood pressure, respiratory rate, and a list of symptoms including cough, sputum, chest pain, muscle pain, abdominal pain or diarrhea, and dyspnea. Health and medical history data include presence or absence of comorbidities (systemic hypertension, diabetes mellitus, asthma, heart disease, emphysema,

immunodeficiency), and smoker status. Laboratory data include conjugated alanine, bilirubin, total bilirubin, creatine kinase, CRP, ferritin, hemoglobin, LDH, leukocytes, lymphocyte, monocyte, platelet, polynuclear neutrophil, and urea.

**CT scan acquisition**. CT scan data were available for 980 patients representing a total of 506,341 2D images (517 slices per patient on average). Summary statistics for the clinical, biological, and CT scan data are provided in Table 1. Three different models of CT scanners were used: two General Electric CT scanners (Discovery CT750 HD and Optima 660 GE Medical Systems, Milwaukee, USA) and a Siemens CT scanner (Somatom Drive; Siemens Medical Solutions, Forchheim, Germany). All patients were scanned in a supine position during breath-holding at full inspiration. The acquisition and reconstruction parameters were of 120 kV tube voltage with automatic tube current modulation (100–350 mAs), 1 mm slice thickness without interslice gap, using filtered-back-projection (FBP) reconstruction (SOMATOM Drive) or blended FBP/iterative reconstruction (Discovery or Optima). Axial images with slice thickness of 1 mm were used for coronal and sagittal reconstructions.

**Radiology reports**. COVID-19-associated CT imaging features were obtained from radiologist reports that follow the guidelines of several scientific societies of radiology (French SFR, STR, ACR, RSNA) regarding the reporting of chest CT findings related to COVID-19[54]. The template of the radiologist report (https://ebulletin.radiologie.fr/actualites-covid-19/compte-rendu-tdm-thoracique-iv-0) was accessed on 17 March and the reports were completed retrospectively for the patients who were admitted to the hospital before that date. CT imaging characteristics were evaluated to provide the following five variables: (i) GGO (rounded/nonrounded/absent) that is defined as an increase in lung density not sufficient to obscure vessels or preservation of bronchial and vascular margins, (ii) consolidation (rounded/nonrounded/absent) that occurs when parenchymal opacification is dense enough to obscure the vessels' margins and airway walls and other parenchymal structures, (iii) the crazy-paving pattern (present/absent) that is defined as ground glass opacification with associated interlobular septal thickening[55], (iv) peripheral topography (present/absent) that corresponds to the spatial distribution of lesions in the one-third external part of the lung, and (v) inferior predominance (present/absent) that is defined as a predominance of lesions located in the lower segments of the lung. A rounded pattern (for GGO and consolidation) is defined as a lesion presenting a well delineated shape. In addition to the five CT imaging features, radiologists assessed the extent of lung lesions according to the evaluation criteria established by the French Society of Radiology (SFR)[56]. Disease extent can be: absent/minimal (<10%)/moderate (10–25%)/extensive (25–50%)/severe (>50%)/critical >75%. The coding absent/minimal/moderate extensive/severe/critical was based on a quantitative variable with values of 0/1/2/3/4/5. Variables were automatically extracted from the report using optical character recognition.

**Statistical analysis**. When detecting association with the severity outcome, odds ratio and p-values (two-sided tests) were computed separately for each hospital using logistic regression (glm function of the R statistical software). p-values from the two different hospitals were pooled using the Stouffer meta-analysis formula accounting for the two different sample sizes. For association between severity and each variable, we considered Bonferroni correction accounting for 58 variables. To compute confidence intervals for AUC values, we considered DeLong method[57]. Survival functions were obtained using Kaplan–Meier estimators. For computing calibration slope and intercept, we considered the rms R package that transforms predicted probabilities to log odds ratios, which are then used as a dependent variable in a logistic regression.

**Deep learning models for severity classification based of CT scans**. The deep learning model was defined as an ensemble of two submodels, as illustrated in Supplementary Fig. 2. Each submodel predicted disease severity from CT scans without using any expert annotations at the slice level. Preprocessing of the data consisted of resizing the CT scans to a fixed pixel spacing of (0.7 mm, 0.7 mm, 10 mm) and applying a specific windowing on the HU intensities. Each submodel is composed of two blocks: a deep neural network called feature extractor and a penalized logistic regression. The two submodels feature extractors are EfficientNet-B0[58] pre-trained on the ImageNet public database and ResNet50[59] pre-trained with MoCo v2[60] on one million CT scan slices from both Deep Lesion[61] and LIDC[62]. Each of these networks provide an embedding of the slices of the input CT scans into a lower-dimensional feature space (1280 for EfficientNet-B0 and 2048 for ResNet50). For the ResNet50-based submodel, we reduced the dimension of the feature space using a principal component analysis with 40 components before applying logistic regression. A different windowing was applied on the CT scans before the feature extractor: (–1000 HU, 600 HU) for EfficientNet-B0 and (–1000 HU, 0 HU), (0 HU, 1000 HU) and (–1000 HU, 4000 HU) for ResNet50. Predictions of AI-severity were obtained by averaging predictions of the submodels using equal weights. Optimization of the architecture of the network (preprocessing, feature extraction or model architecture and training, feature engineering, model aggregation) was performed using a fivefold cross-validation on the training set of 646 patients from KB.

CT scans may contain devices such as catheters (EKG monitoring, oxygenation tubing, etc.) that are easily detectable in a CT and can bias prediction of severity. Indeed, there is a risk of detecting the presence of a technical device associated with severity instead of detecting the radiological features associated with severity[63]. In order to ensure that medical devices do not affect feature extraction, all voxels outside of the lungs were masked using a pre-trained U-Net lung segmentation algorithm[64].

**Multivariate models to predict severity**. The different models that combine multiple features to predict severity were fitted using logistic regression (AI-severity, AI-segment, C & B, C & B & RR). Models were trained using cross-validation with five folds on the training dataset of 646 patients from KB, and folds were stratified by age and severity outcome. Variables that were available for less than 300 patients of the training set (conjugated bilirubin and alanine) were not used. For the remaining variables, missing values were imputed by the average over patients of the training set. L2 regularization was applied when fitting logistic regression. The regularization coefficient was determined by maximizing the average AUC over the five cross-validation folds, using a range of different values ranging from 0.01 to 100. XGBoost algorithm was also evaluated but did not show better performance than logistic regression. We use pandas and scikit-learn to manipulate data, train and evaluate machine learning algorithms[65].

To select variables in the multivariate models, we considered a forward feature selection technique (Supplementary Fig. 8). The first variable included in the model is the variable that provides the largest AUC values. Then, we computed AUC values for all models with two variables including the first one that has already been included. We continued this procedure until all variables were included. Performances of the models increased quickly when the first variables were included and then AUC values reached a plateau (Supplementary Fig. 8). We used the elbow method to select the parsimonious set of variables that is found when a plateau of AUC is reached. For the three models that include CT scan information, we consider the model C & B & RR to perform variable selection. The three models (AI-severity, C & B & RR, AI-segment) were then trained using the six variables found with our variable selection procedure. The variable selection procedure for the C & B model that contains clinical and biological variables only indicated that ten variables should be kept in the model (Supplementary Table 5).

**Other scores to predict severity and mortality**. We performed a comparison of the proposed AI-severity model with 11 other COVID-19 severity scores published in the literature: COVID-Gram[6], two scores from Colombi et al.[16], CALL[9], CURB-65[66], Yan et al.[7], Liang et al.[67], NEWS2 and NEWS2 for COVID-19[68], 4C mortality score[12], and MIT analytics (https://www.covidanalytics.io/mortality_calculator). Among these scores, only three included radiological information in their model: the presence of an X-ray abnormality for COVID-Gram and Liang et al., and the lung disease extent for Colombi et al. The number of considered clinical and biological variables for these scores varied (from 3 for Yan et al. to 14 for NEWS2 for COVID), as well as the model architecture (simple scoring system, logistic regression, XGBoost, or multilayer perceptron), and the outcome they were trained on (such as death or admission to ICU). Notable variation between scores includes the definition of comorbidities and details about how it has been computed for the different scores are provided in Supplementary Table 6. For missing variables, we manually imputed the missing variables with a constant value (Supplementary Table 6). Due to the poor performances of one of the score[7], we retrained their score by repeating their training procedure with the KB development cohort.

**Reporting summary**. Further information on research design is available in the Nature Research Reporting Summary linked to this article.

## Data availability

The dataset of patients hospitalized at Kremlin-Bicêtre (KB) and Institut Gustave Roussy (IGR) are stored on a server at Institut Gustave Roussy (IGR). The data are available from the first author upon request subject to ethical review.

## Code availability

Code to execute AI-severity as well as the other models we developed and the 11 additional models we evaluated are available online at https://github.com/owkin/scancovia.

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

## Acknowledgements

We would like to thank J.-Y. Berthou, H. Berry, and P. Gesnouin from Inria, D. Debenedetti, M. He, R. Patel, G. Rouzaud, B. Schmauch, and J. Du Terrail from Owkin, and F. Lion from Gustave Roussy for their support. This work was granted COVID-19 priority access to the HPC resources of IDRIS (Jean Zay) under GENCI allocations 2020-AD011011728 and 2020-AD011011769.

## Author contributions

N. L., S. A., E. C., P. H., R. D., N. L., P. T., E. B., M. S., A. S., F. C., S. J., M. S., I. B., J. D., J.-C. P., H. T., E. P., G. W., T. C., F. B., M.-F. B., M.B. conceived the idea of this paper. N. L., S. A., E. C., H. G., P. H., M. D., S. S., O. M., M.-P. T., J.-P. L., R. M., N. L., P. T., E. B., G. G., C. B., S. J., F. G., N. T., Y. L., T. D., K. G., A. N., M. T., S. V., M. S., I. B., Y. B., E. P., M. A., J. D., F. B., A. G., J. D., J.-C. P., H. T., E. P., G. W., T. C., F. B., M.-F. B., M. G. B. B. participated to the acquisition and treatment of data. N. L., S. A., E. C., P. H., R. M., N. L., P. T., E. B., S. J., M. S., P. J., I. B., J. D., J.-C. P., H. T., E. P., G. W., T. C., M.-F. B., M. G. B. B. implemented the analysis. N. L., S. A., E. C., P. H., R. M., N. L., P. T., E. B., S. J., M. S., I. B., J. D., J.-C. P., H. T., E. P., G. W., T. C., M.-F. B., M. G. B. B. contributed to the writing of the manuscript.

## Competing interests

The authors declare the following competing interests: Employment: M. G. B. B., P. H., R. D., N. L., P. T., E. B., S. J., M. S., P. J., F. B., O. D., J.-B. S., K. S., E. P., J. D., A. G., employed by Owkin. Co-founders of Owkin Inc: T. C., G. W. The remaining authors declare no competing interests.
