## [Peer Review File · Nature Communications]

Reviewers' Comments:

Reviewer #1:

Remarks to the Author:

The authors present a methodology to predict COVID-19 patient outcomes through an integration of clinical characteristics, lab tests, and chest CTs. They propose two machine learning pipelines and combine them with clinical characteristics and test their prognostic capabilities through three different outcomes, a high-severity, a combined outcome (death or ICU admission), and death outcome. The retrospective tests were performed on a subset of the two hospitals included in the analysis. Their tests show that in this specific cohort, the models that combine one of the proposed ML pipelines, along with the clinical and biological data, most of the times, display better AUCs than the individual components alone, or other proposed models in the literature.

I find the initial premise of the study exciting and very timely. I agree with the authors that combining radiological measurements with biological variables and other patient information can greatly enhance the prognostic and diagnostic value of AI algorithms in this and other diseases. However, I was underwhelmed by both the results presented and the way they were presented. The manuscript that I reviewed did not resemble a scientific paper. The manuscript is hard to read, confusing and in some cases not clear whether it's a scientific paper or a progress report of a project. While some parts of the paper are relatively well written, other parts seem hastily written. The methods seem to be all thrown in the supplementary material, there is a hint of an introduction and a complete absence of discussion. The manuscript was extremely hard to review due to the continuous back and forth that I had to do between the main article and the supplementary material, on top of the fact that a lot of the methodological steps having unclear points. Beyond these serious editorial issues that demand a copy editing overhaul to be in publishable shape, the results of the paper were underwhelming and I have some serious reservations on the claims being made in this paper with regards to the superiority and CT scans being the "strongest performance booster". An increase of 0.02 and 0.03 in a test set of 150 and 137 patients cannot be considered significant (and they were not shown to be).

Portability/robustness by testing an algorithm on a small subset of an already rather small number of patients from two hospitals that were already included in the training set cannot be claimed. Finally, superiority over other algorithms cannot be claimed when in one out of two cases, a much simpler algorithm that does not use any complicated, time consuming and expensive CT scans (such as the MIT mortality calculator) outperforms all proposed models. Given that the clinical use of the models proposed are of utmost importance to the health of patients and the limited number and ways that they were tested, their value, for now, remains theoretical. Unfortunately, as it stands, the manuscript is not appropriate for publication.

Some specific points on the manuscript.

The study proposed two machine learning pipelines: AI-segment and AI-severity. The methodology applied in the construction of these pipelines is unclear, it may seem that the authors tried some image and 3D processing deep neural networks and then combined them in an ensemble in a usual manner, averaging over the predictions of the submodels. The submodels are well-known and documented in the machine learning literature, however, how they were chosen is unclear from reading the manuscript.

The AI-segment pipeline includes large, deep neural network submodels. It is concerning to see that these segmentation submodels were only trained using a handful of images without pretraining. The pretraining of segmentation deep neural networks is common practice and is highly suggested for the small training set this study works with. The overall pipeline and training procedure of AI-segment looks valid. Submodels of the AI-severity pipeline were again, well-documented and -used deep learning models, pre-trained on Imagenet. Similarly to the AI-segment pipeline, the choice of the submodels is unclear and the authors should elaborate on how they built the ensemble. Although there are more sophisticated methods to replace missing values than taking the average of the given variable, the small number of samples used in the study may prevent the application of more complex methods: e.g. linear or Bayesian prediction of missing values given the available ones.

The authors need to state whether free-text dictation, structured reporting, or fixed picklist

reporting was used to examine the scans and provide examples. Were the reports in English? what were the exact terms used? The template for reading images should be provided in the appendix, as well as a sample report used for OCR and data extraction.

One other point that was confusing is that their readings, although claimed to be prospective, were based on the consensus statement by STR, ACR, and RSNA. The consensus statement was made available online on March 25th. The authors claim that the reads happened prospectively, with the first patient being admitted on February 12th. I have trouble understanding whether these reads were indeed prospective or retrospective.

Minor points:

1. Instead of average (16) annotations, provide min, max, and median
2. For the AI severity model, the authors report features correlating with age and being able to discriminate gender and they need to provide a lot more details on these findings since they only looked at the lungs.

Reviewer #2:

Remarks to the Author:

In this manuscript, the authors developed a CT based AI system to automatically segment the scans and reproduce radiologists' annotations. In addition, the CT features were used together with the clinical data to develop a prognostic model to predict the prognosis of patients with COVID-19.

A similar study using deep learning for CT-based AI model to detect COVID-19, segment and to early predict future severity, titled "Clinically Applicable AI System for Accurate Diagnosis, Quantitative Measurements, and Prognosis of COVID-19 Pneumonia Using Computed Tomography" has already been published ([https://www.cell.com/cell/fulltext/S0092-8674\(20\)30551-1?rss=yes#main-menu](https://www.cell.com/cell/fulltext/S0092-8674(20)30551-1?rss=yes#main-menu)). Compared with the Cell paper, this manuscript used a smaller dataset, and focused more on the severity prediction by using a multi-modal approach. In my opinion, the most valuable contribution of this manuscript is the data cohorts with manually labeled CT images, clinical characteristics and lab tests, which could be an important public resource. If the datasets used in this study can be released to public, it will greatly facilitate the research community to develop CT-based methods for early severity prevention of the COVID-19 infection, as well as improve the treatment of patients. Nevertheless, the AI model could have clinical utility and be an important tool to help combat the COVID-19 pandemic.

Major comments:

1. How do you determine some hyper parameters when training the AI-segment and AI-severity models? In general, the data should be split into training set, validation set and testing set, while it seems that the validation set is omitted in the study.
2. The authors trained a deep neural network called AI-segment to segment radiological patterns and provide automatic quantification. However, the dataset for this kind of learning-based method (161 patients) seems small, which makes it difficult to interpret evaluation results.
3. To assess disease progression and build model to stratify risk, the authors measured the time from admission to critical severity, or death. However, there is no mention of when the CT was taken during the hospital course. Could you provide this information? Were all the chest CTs were obtained on admission? Failure to account for time-of-CT in the analysis could result in a lead time bias and data gathered at time of the clinical endpoints could skew the results for patients scanned earlier in the disease course.
4. How do the authors propose to implement the AI system clinically in the near term? For risk stratification, which physicians will be the target user? Will the main goal be to aid in radiologists with less experience? Is this emergency room physicians, or pulmonologists? Again, more concretely how do the authors propose to integrate the AI system into clinical practice workflow in the near term?

Minor comments:

1. The statistics used should be defined in the paper.
2. The authors used AUC to evaluate the performance of the classification model. The 95% confidence interval should be reported.
3. For the evaluation of AI-segment models, the commonly used metrics were not presented in the paper. Could you refer to the DICE coefficient to show a threshold-free metric, which may better evaluate the performance of the segmentation model.
4. Figure 3 is a typo. "UCI" -> "ICU"?
5. For the AI-severity model, have you ever tried other aggregation methods except averaging the feature vector?
6. The manuscript needs to be polished.

We would like to thank the associate editor and the reviewers for their comments about our manuscript. We have now considerably changed the text of the manuscript and we have included the Methods section in the main text.

The main addition to the manuscript is a thorough comparison with six different risk scores for COVID severity (Figure 3) stressing the added value of our proposed risk scores that include CT-scan information. We have re-trained our prognosis scores and confirmed that they are more predictive of severity when including CT-scan data.

Beyond AI modeling, we now provide a simple 6-variable severity *ScanCov* score integrating a radiological quantification of lesion extent with key clinical and biological variables. The *ScanCov* score provides more accurate predictions than published COVID severity scores (Figure 3), and can rapidly become a reference scoring approach for severity prediction.

Once the paper is published, we will release our source code in a public github repository (<https://github.com/owkin/scancovia>) and an online calculator will be available to compute the ScanCov score in a clinical context.

The main changes in the manuscript are highlighted in light yellow.

Michael Blum, on behalf of the authors

Reviewer #1 (Remarks to the Author):

The authors present a methodology to predict COVID-19 patient outcomes through an integration of clinical characteristics, lab tests, and chest CTs. They propose two machine learning pipelines and combine them with clinical characteristics and test their prognostic capabilities through three different outcomes, a high-severity, a combined outcome (death or ICU admission), and death outcome. The retrospective tests were performed on a subset of the two hospitals included in the analysis. Their tests show that in this specific cohort, the models that combine one of the proposed ML pipelines, along with the clinical and biological data, most of the times, display better AUCs than the individual components alone, or other proposed models in the literature.

I find the initial premise of the study exciting and very timely. I agree with the authors that combining radiological measurements with biological variables and other patient information can greatly enhance the prognostic and diagnostic value of AI algorithms in this and other diseases. However, I was underwhelmed by both the results presented and the way they were presented.

Comment

The manuscript that I reviewed did not resemble a scientific paper. The manuscript is hard to read, confusing and in some cases not clear whether it's a scientific paper or a progress report of a project. While some parts of the paper are relatively well written, other parts seem hastily written. The methods seem to be all thrown in the supplementary material, there is a hint of an introduction and a complete absence of discussion. The manuscript was extremely

hard to review due to the continuous back and forth that I had to do between the main article and the supplementary material, on top of the fact that a lot of the methodological steps having unclear points.

Our answer

We have now completely revised the text of manuscript as recommended. We have moved the text of the supplementary material to the Methods section of the main paper and the corresponding text has been heavily modified. We have also supplemented the Introduction by additional paragraphs and added a Discussion section.

Comment

Beyond these serious editorial issues that demand a copy editing overhaul to be in publishable shape, the results of the paper were underwhelming and I have some serious reservations on the claims being made in this paper with regards to the superiority and CT scans being the “strongest performance booster”. An increase of 0.02 and 0.03 in a test set of 150 and 137 patients cannot be considered significant.

Our Answer

We have updated our analysis of the added value of CT-scan information.

1. For each outcome considered and validation set, both *ScanCov* and *AI-severity* performed better than the bimodal biological/clinical *C & B* model (see Figures 2 & 3). The gain of performance when compared to the *C & B* model was larger for the KB dataset (median AUC increase of 4.0% for *AI-severity* and 3.6% for *ScanCov*) than for the IGR dataset (median AUC increase of 1.5% for *AI-severity* and 0.4% for *ScanCov*).
2. Among all the additional scores we evaluated, we found that the COVID-GRAM is the score with the largest AUC and it is also the only alternative score that includes CT-scan information.
3. We further investigated the radiology disease extent feature to confirm that it brings additional prognostic information that is not otherwise captured in any clinical or biological variable.. We investigated its relationship with the other variables using the larger KB dataset (Supp Table 6). The 3 variables that were the most correlated with disease extent are LDH ($r = 0.52$, 95% C.I. = (0.45,0.58)), CRP ($r = 0.45$, 95% C.I. = (0.39-0.51)), and oxygen saturation ($r = -0.43$, 95% C.I. = (-0.49,-0.37)). We then regressed the severity outcome with disease extent and the three correlated variables and found that significant predictors included oxygen saturation ($P = 1.57e-07$) and disease extent ($P = 0.01$), whereas statistical evidence for association was weak for LDH ($P = 0.06$) and absent for CRP ($P = 0.26$). The statistical evidence for association between disease extent and severity was also found ($P = 9.85e-08$) when accounting for the five additional variables of the *ScanCov* score.

This being said, we acknowledge that the writing in the first version of the paper was not balanced enough. In the current version of the manuscript, we acknowledge that there is information **redundancy between** CT-scan, biological and clinical measures. The main reason is that the extent of lesions as evaluated on a CT scan is correlated with biological markers of inflammation (CRP), tissue damage (LDH) and clinical measure of oxygenation.

Comment

Portability/robustness by testing an algorithm on a small subset of an already rather small number of patients from two hospitals that were already included in the training set cannot be claimed.

Our answer

Only KB patients were included in the training set, not any IGR patients. This is explicitly written in the text “All three models were trained on 646 KB patients, tested on 150 KB validation patients, and validated on the independent IGR dataset of 135 patients”. Our cohort is of the same order of magnitude than other cohorts used to develop prognosis score. The prognosis score of Zhang et al. (2020, DOI: 10.1016/j.cell.2020.04.045) was based on 456 hospitalized patients with clinical outcome information. The development cohort of the COVID-GRAM score included 1590 patients (doi:10.1001/jamainternmed.2020.2033), and the cohort of the CALL score included 208 patients (<https://doi.org/10.1093/cid/ciaa414>).

Comment

Finally, superiority over other algorithms cannot be claimed when in one out of two cases, a much simpler algorithm that does not use any complicated, time consuming and expensive CT scans (such as the MIT mortality calculator) outperforms all proposed models. Given that the clinical use of the models proposed are of utmost importance to the health of patients and the limited number and ways that they were tested, their value, for now, remains theoretical.

Our answer

We have now included a more thorough comparison with existing scores that predict COVID severity (see new Figure 3). There is a new section in the Results section describing this comparison. The new text reads as follows

“The models *ScanCov* and *AI-severity* also outperformed other severity or mortality scores (Figure 3, Supp Fig 7, Supp Table 3). The median difference (averaged over outcomes) between the AUC of *AI-severity* and of other scores ranged between 5% (COVID-GRAM) and 15% (CALL) at KB and between 10% (COVID-GRAM) and 26%¹⁵ at IGR. The median difference (averaged over outcomes) between the AUC of *AI-segment* and of other scores ranged between 2% (COVID-GRAM) and 12% (CALL) at KB and between 5% (COVID-GRAM) and 24%¹⁵ at IGR. Similarly, the median difference (averaged over outcomes) between the AUC of *ScanCov* and of other scores ranged between 4% (COVID-GRAM) and 14% (CALL) at KB and between 5% (COVID-GRAM) and 24%¹⁵ at IGR.”

In addition to the AI scores based on deep learning models, we also provide a simple 6-variable score that uses radiologist evaluation of disease extent as CT scan information. The simple score is superior to the CALL score, to the MIT analytics score, to the score of Colombi & al.(2020), to the score of Yan et al. (2020). It is also superior to the COVID-GRAM score although the difference is less pronounced because the COVID-GRAM score also includes CT-scan information. The reviewer mentions the MIT mortality calculator, which has indeed the best performance to predict death on the validation KB dataset.

However, its performance is less robust than the ScanCov score and it is for instance one of the worst model to predict death on the IGR dataset.

Some specific points on the manuscript.

Comment

The study proposed two machine learning pipelines: AI-segment and AI-severity. The methodology applied in the construction of these pipelines is unclear, it may seem that the authors tried some image and 3D processing deep neural networks and then combined them in an ensemble in a usual manner, averaging over the predictions of the submodels. The submodels are well-known and documented in the machine learning literature, however, how they were chosen is unclear from reading the manuscript.

Our answer

The manuscript explores 2 machine learning pipelines.

- The AI-segment pipeline aims to precisely quantify the presence of different lesions in the lungs in contrast to the semi-quantitative quantification performed by radiologists.
- The AI-severity pipeline aims to directly predict the associated severity, without the use of radiologist annotations.

The description of both pipelines can be found in the Methods section under the subheadings “Machine learning models for segmentation of CT scans (AI-segment)” and “Machine learning models for severity classification based of CT scans (AI-severity)”.

For the AI-segment model, we have investigated several state-of-the-art neural network architectures for image segmentation, namely 2.5D UNet, UNet, Residual UNet, Dense UNet, 3D ResNet (with 10, 18, 34, 50, 101, 152, or 200 layers), 3D UNet, and 3D Residual UNet. Each architecture was evaluated using a 5 fold cross-validation scheme on the training set. The 2D UNet showed its superiority for lung segmentation. When segmenting the lesions, we opted for 2.5D UNet and 3D ResNet with 50 layers by processing 2.5D and 3D representations of the scans. This approach presents the advantage of combining efficiently both partial annotations and full annotations. Performing ensemble on the two retained models is efficient in terms of performance as well as inference time per scan and memory usage.

For the AI-severity model, numerous architectures have been explored. Each architecture was evaluated using a 5 fold cross validation scheme on the training set. Among the explored architectures we can cite: alternative pretrained backbones, alternative methods for pooling such as attention models, auxiliary loss using clinical variables, feature engineering on the backbone features or the aggregation method. As most of these variants only added little or no performance boost, we decided to choose the most simple architecture to avoid overfitting on the training set. We do not mention exploration of the different architectures in the manuscript for sake of clarity.

Comment

The AI-segment pipeline includes large, deep neural network submodels. It is concerning to see that these segmentation submodels were only trained using a handful of images without pretraining. The pretraining of segmentation deep neural networks is common practice and is highly suggested for the small training set this study works with. The overall pipeline and training procedure of AI-segment looks valid.

Our answer

We agree with the reviewer's point that neural network submodules in the *AI-segment* model are large and may be difficult to train on small datasets. To be precise, 2.5D Unet has 13 million parameters and 3D ResNet50 has 40 million parameters. However, our fully annotated scans (FAS) training set has in total 3704 512x512 slices resulting in a sample of 1 billion pixels. Following the reviewer's suggestion, we have now trained the networks by initializing weights with pretrained weights, which indeed was beneficial in terms of segmentation accuracy. However, this did not have a major impact in the estimated volumetric ratios, and thus did not change the prognosis performance of *AI-segment*.

Comment

Submodels of the AI-severity pipeline were again, well-documented and -used deep learning models, pre-trained on Imagenet. Similarly to the AI-segment pipeline, the choice of the submodels is unclear and the authors should elaborate on how they built the ensemble.

Our answer

The submodels (EfficientNet and ResNet) were selected looking at their performances on the training set using a 5 fold cross validation scheme. We performed a linear aggregation of the submodels predictions and found through grid search that simple average provided close to the best performances. During the revision of the paper we found that having 2 different intensity preprocessing for both EfficientNet and ResNet did not improve performance and thus we decided to keep only one intensity preprocessing (1 for EfficientNet and 1 for ResNet), further simplifying the pipeline. We also added a preprocessing step to normalize slice resolution to 0.7mm / pixels .The EfficientNet model is pretrained on ImageNet while the ResNet model is pretrained using a very recent self supervised technique (MoCo v2, Chen et al. 2020 arXiv:2003.04297) on more than a million CT scan slices from public datasets.

Comment

Although there are more sophisticated methods to replace missing values than taking the average of the given variable, the small number of samples used in the study may prevent the application of more complex methods: e.g. linear or Bayesian prediction of missing values given the available ones.

Our answer

We agree with the reviewer comment that there are more sophisticated methods to replace missing values. However, for sake of simplicity, we decided to keep our simple average method.

Comment

The authors need to state whether free-text dictation, structured reporting, or fixed picklist reporting was used to examine the scans and provide examples. Were the reports in

English? what were the exact terms used? The template for reading images should be provided in the appendix, as well as a sample report used for OCR and data extraction.

Our answer

It is a structured report in French which has been provided by the “Société Française de Radiologie” (French Radiology Society). The link to the template is now provided in the METHODS section.

Comment

One other point that was confusing is that their readings, although claimed to be prospective, were based on the consensus statement by STR, ACR, and RSNA. The consensus statement was made available online on March 25th. The authors claim that the reads happened prospectively, with the first patient being admitted on February 12th. I have trouble understanding whether these reads were indeed prospective or retrospective.

Our answer

We understand the confusion and we now write in the main text the following sentence “The template of the radiologist report was available the 17th of March and the reports were completed retrospectively for the patients who were admitted to the hospital before this date.”.

Minor points:

Comment

1. Instead of average (16) annotations, provide min, max, and median

Our answer

We thank the reviewer for this suggestion and now provide in the subsection ‘Segmentation of CT-scans’ the median, minimum and maximum values of the segmentation error of each lesion type, on the fully annotated scans.

Comment

2. For the AI severity model, the authors report features correlating with age and being able to discriminate gender and they need to provide a lot more details on these findings since they only looked at the lungs.

Our answer

We provided more detailed results on the ability for the (updated) AI-severity model to predict clinical and radiological variables in the supplementary table 5. We hypothesize that a large part of the capability of the model to predict gender and age comes from its ability to estimate variables correlated to them. For instance, we computed an AUC of 75% when comparing lung volume, computed from the segmentation masks and gender.

Reviewer #2 (Remarks to the Author):

In this manuscript, the authors developed a CT based AI system to automatically segment the scans and reproduce radiologists' annotations. In addition, the CT features were used

together with the clinical data to develop a prognostic model to predict the prognosis of patients with COVID-19.

Comment

A similar study using deep learning for CT-based AI model to detect COVID-19, segment and to early predict future severity, titled "Clinically Applicable AI System for Accurate Diagnosis, Quantitative Measurements, and Prognosis of COVID-19 Pneumonia Using Computed Tomography" has already been published ([https://www.cell.com/cell/fulltext/S0092-8674\(20\)30551-1?rss=yes#main-menu](https://www.cell.com/cell/fulltext/S0092-8674(20)30551-1?rss=yes#main-menu)). Compared with the Cell paper, this manuscript used a smaller dataset, and focused more on the severity prediction by using a multi-modal approach.

Our answer

We agree that there are similarities with the Cell paper. However, there are important differences: (i) our paper focuses on prognosis whereas the Cell paper mostly focused on diagnosis, (ii) we have a more thorough analysis of how CT scan adds information for prognosis by considering several approaches including radiologist quantification and AI systems, and (iii) we have now included a comprehensive comparison with existing severity prediction scores. In terms of sample size, the cell paper analyses a cohort of 4,154 patients ($\frac{1}{3}$ of COVID patients, $\frac{1}{3}$ of non-sick individuals, and $\frac{1}{3}$ of individuals with other pneumonia) for which 617,775 images have been obtained, but the prognosis analysis concerns only a subset of the data consisting of 456 hospitalized patients. Our paper considers a cohort of 1,000 patients (COVID patients only) for which 418,000 images have been obtained.

Comment

In my opinion, the most valuable contribution of this manuscript is the data cohorts with manually labeled CT images, clinical characteristics and lab tests, which could be an important public resource. If the datasets used in this study can be released to public, it will greatly facilitate the research community to develop CT-based methods for early severity prevention of the COVID-19 infection, as well as improve the treatment of patients. Nevertheless, the AI model could have clinical utility and be an important tool to help combat the COVID-19 pandemic.

Our answer

We thank the reviewer for acknowledging that the data cohort is valuable. However, we will not be able to make the data public. The consent form signed by the patients does not give us the opportunity to make the data public. We sincerely hope that the numerical summaries provided in Figure 1 can be valuable for researchers for instance when doing a meta analysis of severity for hospitalized patients. Data can be available upon request from the first author. The Data Availability section reads as "The dataset of patients hospitalized at Kremlin-Bicêtre (KB) and Institut Gustave Roussy (IGR) are stored on a server at Institut Gustave Roussy (IGR). The data are available from the first author upon request subject to ethical review."

Major comments:

Comment

1. How do you determine some hyper parameters when training the AI-segment and AI-severity models? In general, the data should be split into training set, validation set and testing set, while it seems that the validation set is omitted in the study.

Our answer: The hyperparameters of our *AI-segment* model are the learning rate and the batch size of the networks. We have carefully described the setting of those hyperparameters in the revised version of the paper. We followed the same strategy for setting those hyperparameters, as in previous works where such neural networks models have been used, and if necessary, fine-tuned on few trials.

For *AI severity*, the hyper parameters search was done by optimizing performances on the training set using a 5 fold cross validation scheme. 5 fold CV provides a stronger evaluation on the training set than a single simple training / validation split, especially because of the size of the training cohort. This has now been mentioned in the manuscript as “Optimisation of the architecture of the network (preprocessing, feature extraction, feature engineering, model aggregation) was performed using a 5 fold cross validation on the training set.”

Comment

2. The authors trained a deep neural network called AI-segment to segment radiological patterns and provide automatic quantification. However, the dataset for this kind of learning-based method (161 patients) seems small, which makes it difficult to interpret evaluation results.

Our answer: The training dataset is indeed of relatively small size essentially due to the difficulty and time necessary to annotate that data, in a crisis period, by much needed healthcare specialists. Following the advice of the reviewer, we have redone our experiments, initializing the networks with pretrained weights from medical imaging public datasets, and we also generated additional annotations. This slightly improved segmentation accuracy, though it did not affect performance of the final severity prediction scores. We have modified the description about our training methodology in the paper accordingly.

Comment

3. To assess disease progression and build model to stratify risk, the authors measured the time from admission to critical severity, or death. However, there is no mention of when the CT was taken during the hospital course. Could you provide this information? Were all the chest CTs were obtained on admission? Failure to account for time-of-CT in the analysis could result in a lead time bias and data gathered at time of the clinical endpoints could skew the results for patients scanned earlier in the disease course.

Our answer

Yes, all CT scans have been obtained in the 48 hours following admission. It is now written explicitly in the main text.

Comment

4. How do the authors propose to implement the AI system clinically in the near term? For risk stratification, which physicians will be the target user? Will the main goal be to aid in radiologists with less experience? Is this emergency room physicians, or pulmonologists? Again, more concretely how do the authors propose to integrate the AI system into clinical practice workflow in the near term?

Our answer

Both the AI systems and the simplest ScanCov score should be provided to the physicians in charge of triage of COVID patients. Some efforts need to be deployed for the AI system to be used in a clinical context and the authors are currently working on that. The code for AI systems will be made available on a public github repository after acceptance of the paper. The ScanCov score, which might be more convenient to use than AI system because it is based on a radiologist quantification of lesion extent instead of a AI quantification, will be provided after acceptance of the paper both on the public github repository and in an online calculator (Calculate by QxMD website, <https://qxmd.com/calculate-by-qxmd>).

Minor comments:

Comment

1. The statistics used should be defined in the paper.

Our answer

We have included a new section in the manuscript entitled “Statistical Analysis”.

Comment

2. The authors used AUC to evaluate the performance of the classification model. The 95% confidence interval should be reported.

Our answer

The confidence intervals are reported in the table 3 of the Supplementary and they are now provided in the main text as well.

Comment

3. For the evaluation of AI-segment models, the commonly used metrics were not presented in the paper. Could you refer to the DICE coefficient to show a threshold-free metric, which may better evaluate the performance of the segmentation model.

Our answer:

Following the reviewer’s comment, we now report, in the revised version of the ms, the F1 scores (using micro-averaging) for the PAS (partially annotated scans) and FAS (fully annotated scans), of the IGR test set, when discriminating lesions and sane lung regions. We also reported the accuracy for background class segmentation, on FAS (no background class was annotated in PAS). Those metrics were preferred to the DICE score (corresponding to F1 score using macro averaging) as they are less sensitive to class imbalance.

In the revised version of the paper, we provide the relative errors (median [min-max]) of volume prediction, for each class, for the FAS, which again illustrates the good ability of *AI-segment* to estimate volume of lesions.

We also provide in supp tab 1 the F1 scores (in addition to accuracy) about detection of a lesion type. To compute F1 score, we considered that the answers available in the radiologist report corresponded to the ground truth.

Comment

4. Figure 3 is a typo. "UCI" -> "ICU" ?

Our answer

Thank you, it has been changed.

Comment

5. For the AI-severity model, have you ever tried other aggregation methods except averaging the feature vector?

Our answer

In the AI-severity model, the predictions of each submodel were averaged. We performed a grid search on the weights for the linear aggregation and found that averaging predictions was the best option. We also performed an additional experiment where we concatenated feature vectors from different models. This experiment did not provide a large enough improvement to be retained in the final manuscript.

Comment

6. The manuscript needs to be polished.

Our answer

Following your comment and the comment of reviewer 1, we have modified and extensively polished the manuscript. Main changes are highlighted in yellow.

Reviewers' Comments:

Reviewer #1:

Remarks to the Author:

I'd like to thank the authors for their responses to my comments and the extensive editing done to the manuscript, which I believe improved the structure and made it easier to review as a scientific paper. However, my two main problems remain in this revised version, mainly the focus of the paper to one, three or more than three models, and the main claim of the paper in the title and throughout the text that inclusion of CT-scan information provides an improvement on a prognostic model (an improvement from what is not necessarily clear).

1. Although the revised manuscript shows in detail the comparison between previous models and the proposed ones, it still remains unclear which model, out of the Tri AI-segment, Tri AI-severity, and Tri RR is the final suggested model to use in each use-case. Authors should clearly explain whether they propose one, or all three models at different use-cases. If they propose one, they should justify why they would propose that one over the others. If the authors propose multiple models they should point out when should the user employ one over the other, justified by superior positive or negative predictive values compared to the others.

2. One of the main claims of the study that the inclusion of CT-scans, or radiology reports derived from CT-scans, improves the performance of their models providing "added prognostic value" (title, abstract, and page 14, lines 11-12). When looking at the ROC AUC of C&B compared to Tri AI-segment, Tri AI-severity, and Tri RR, we only see a 4% improvement at best (Figure 2); one would generously claim this is a marginal improvement, especially for such small samples. The overlapping confidence intervals in Figure 3 similarly hints at such a non-significant AUC improvement and strongly questions whether that marginal superiority would hold once the models are implemented in clinical practice and validated prospectively or externally. Indeed, a 4% improvement when evaluated on a small test set of 150 and 135 patients, hardly have any statistical significance and does not hold with the different shuffling of the data (as is evident from Figure 3).

Hence, unless the authors show a statistically significant difference of AUCs between C&B and any of the Tri models, the claims of superiority or improvement should be removed. The title should be rephrased, because it's not clear which part of "clinical characteristics, lab tests and chest CTs" actually improve the model performance against which baseline model. Furthermore, the title and claims in the text also allude to model improvement due to the incorporation of CT-scan information over clinical characteristics and lab tests, which as mentioned above, is doubtful and should be supported by statistical tests. Other changes in the text might further be necessary in case the statistical significance does not hold.

Moreover, beyond the statistical evaluation, there is a serious consideration that the authors should take into account and include a thorough discussion, on the added value of incorporating CT-scans for a truly clinically significant benefit. Incorporating CT-scans into prognostic models implicitly introduces a significant cost to the healthcare system (and possibly the patients) and harmful radiation to the patient. It further requires time from radiologists and lab technicians to collect these scans, as well as machine learning engineers, radiologists to label these images, and the employment of complex machine learning pipelines, which have to be trained on billions of pixels – as pointed out in the answers. In light of all the costs associated with the incorporation of CT-scans, one would expect a very significant improvement in prognostic value.

I am pleased to see that taking pre-trained models in the AI-segment pipeline benefited segmentation accuracy, even if the prognostic value did not increase significantly. Although using a pre-trained base model most probably eases the requirement for a larger training dataset, I would still like to point out that the number of pixels or one could mention the number of bytes a dataset holds, is not a meaningful representation of the amount of useful information said data contains; at the end of the day, the model was trained on 22 fully annotated scans, seeing limited

variations between lungs of different patients. This could be a reason for the marginal prognostic improvement in the performance of the AI-segment model that obviously pre-training could not help with, as the pre-training datasets usually don't include lung CTs.

I still believe the authors have a well-performed ML study on CT scans related to COVID-19. However, it doesn't seem that their claims are backed up by their results. Maybe the paper could present these results in a clearer manner, that in this limited dataset, the inclusion of CT scans does not significantly improve a model that performs relatively well in internal, retrospective validation, using only clinical characteristics and lab tests. I'd leave it to the editors to decide if such a paper would be publishable on their journal, although I would argue that sometimes papers that publish non-results can be as informative as one presenting a positive result.

Reviewer #2:

Remarks to the Author:

In this revised manuscript, Dr Blum and colleagues have revised the paper in line with reviewers' comments. While the article included more network setting description and statistical analysis, the paper continues to be a bit difficult to follow and confusing in places. Moreover, a number of major issues remain.

Major Issues

1. Originality: The methodology presented in this work is not novel and based on the existing advances in machine learning.
2. One previously raised question was whether the AI severity model be able to predict clinical metadata such as age and gender only with lung features. The authors reported "AUC of 0.88 for predicting an age larger than 60 year-old". Rather than AI could learn "to estimate variables correlated to age/" from lung directly, my hypothesis is what AI learns is the natural distribution of age of the data. As the elderly older than 60 years constitute the majority of COVID-19 patients. Information about this should be addressed in the revised manuscript if possible.
3. The accuracy of lung lesion segmentation is relatively low in comparison to the results reported in the literature. A possible explanation could be the fact that the utilized CT slices are very pathological. It is, however, not clear is such moderate-quality segmentations can help with the following analysis such as prognosis.
4. For the evaluation of the AI-segment model, the authors adopted several threshold-based methods in the revised manuscript. However, there seems no validation set for this task. Also, how did the authors select the thresholds?
5. In the paper, the authors initialized the network with pretrained weights. On the other hand, the setting of the initial learning rate is 0.1 for the 3D resnet and 0.001 for the rest two networks. My concern is that these settings are so large that they may damage to the pretrained model. Have the authors tried to address this issue.
6. The paper as written and structured is a bit difficult to follow throughout. Still need to be polished.

We would like to thank the reviewers for their review and constructive comments about our manuscript. We have now considerably extended our comparative study because we now compare *AI-severity* to 11 scores that have been recently published. It is the most comprehensive comparison of the severity/mortality scores for COVID-19 hospitalized patients to our knowledge. We provide in our github repository (<https://github.com/owkin/scancovia>), code to compute *AI-severity* and the 11 alternative scores. In the current version of the ms, the alternative scores include a score recently published in Nature Communications by Liang et al. (<https://www.nature.com/articles/s41467-020-17280-8>) and the 4C mortality score published in BMJ 3 weeks ago and based on data from around 50,000 patients in UK (<https://www.bmj.com/content/370/bmj.m3339>). For the validation IGR cohort, AUC values of *AI-severity* is significantly larger than the AUC of all other scores except when comparing, for the main severity outcome, *AI-severity* to the NEWS2 for COVID-19 score. For the KB development cohort, *AI-severity* has also larger scores compared to alternatives but depending on outcome, and of the alternative, differences may not be significant (see Figure 2).

We answer below to the reviewers' comments.

REVIEWER COMMENTS

Reviewer #1 (Remarks to the Author):

1. Although the revised manuscript shows in detail the comparison between previous models and the proposed ones, it still remains unclear which model, out of the Tri AI-segment, Tri AI-severity, and Tri RR is the final suggested model to use in each use-case. Authors should clearly explain whether they propose one, or all three models at different use-cases.

Answer

We now present mostly the *AI-severity* score, which is the name of the trimodal score based on the deep learning prognosis model. We acknowledge that there were too many newly developed scores in the former version of the ms which was a possible source of confusion for the reader. *AI-severity* is the score that is put forward in the main text. The two other scores based on the radiologist report and based on automatic segmentation are now mentioned briefly in the main text and have slightly smaller AUC compared to *AI-severity* although differences are not significant.

2. One of the main claims of the study is that the inclusion of CT-scans, or radiology reports derived from CT-scans, improves the performance of their models providing "added prognostic value". Hence, unless the authors show a statistically significant difference of AUCs between C&B and any of the Tri models, the claims of superiority or improvement should be removed.

Answer

Following the suggestion of the reviewer, we now report results about statistical significance, which confirms the significant added value of CT-scan although AUC improvement is modest. We have considerably updated the wording of the ms to attenuate previous claims.

We have modified our claims with respect to the large boost followed by including CT-scan information. We now have computed and reported P-values when comparing the trimodal score (now named *AI-severity* in the current version of the ms) and the score with biological and clinical information only (Supp Fig 5). For 2 out of 3 outcomes, differences are significant when evaluating differences for the IGR validation cohort but differences are not significant for the KB development cohort. In the results section, we write “This comparative analysis shows that CT-scan adds significant prognosis information although the addition of CT-scan information increases AUC by a measurable but limited amount in both cohorts; there was a difference of AUC of 0.03 when comparing *AI-severity* to the *C & B* score.” In the discussion section, we write “the neural network prognosis variable was correlated to biological and clinical severity biomarkers such as CRP levels, tissue damage (LDH) and oxygenation—highlighting some information redundancy between data modalities—explaining the relatively modest gain of AUC provided by CT-scan”.

3. Moreover, beyond the statistical evaluation, there is a serious consideration that the authors should take into account and include a thorough discussion, on the added value of incorporating CT-scans for a truly clinically significant benefit. Incorporating CT-scans into prognostic models implicitly introduces a significant cost to the healthcare system and harmful radiation to the patient.

Answer

Chest CT is now used nearly systematically at hospital admission to diagnose COVID-19 especially when RT-PCR results are negative (Herpe, Guillaume, et al. "Efficacy of Chest CT for COVID-19 Pneumonia in France." *Radiology* (2020): 202568) and using *AI-severity* would not introduce any additional exams. We now cite this reference and write the following sentence in the introduction “CT-scans can be acquired at admission to diagnose COVID-19 when RT-PCR results are negative²¹”.

4. I am pleased to see that taking pre-trained models in the AI-segment pipeline benefited segmentation accuracy, even if the prognostic value did not increase significantly. Although using a pre-trained base model most probably eases the requirement for a larger training dataset, I would still like to point out that the number of pixels or one could mention the number of bytes a dataset holds, is not a meaningful representation of the amount of useful information said data contains; at the end of the day, the model was trained on 22 fully annotated scans, seeing limited variations between lungs of different patients.

Answer

Methodological description about AI-segment (including the description of the number of pixels) has been moved to a supplementary document and is not described in the main text anymore. We would like to point out that the model includes, in the training of 2.5D U-Net, partially annotated scans from 176 KB patients, on top of the 22 fully annotated scans, so we consider such a dataset is actually representative of the variability among covid-19 patients.

5. I still believe the authors have a well-performed ML study on CT scans related to COVID-19. However, it doesn't seem that their claims are backed up by their results. Maybe the paper could present these results in a clearer manner, that in this limited dataset, the

inclusion of CT scans does not significantly improve a model that performs relatively well in internal, retrospective validation, using only clinical characteristics and lab tests.

Answer

We thank the reviewer for this comment. We have now included a description of statistical significance when comparing performance of models. We acknowledge that improvement provided by CT-scan information is measurable but limited because of redundancy with other markers of severity such as oxygenation, LDH, and CRP markers.

Reviewer #2 (Remarks to the Author):

Major Issues

1. Originality: The methodology presented in this work is not novel and based on the existing advances in machine learning.

Answer

The objective of our study was not to present original advances in machine learning, although we do believe that our use of a weakly-supervised Deep-Learning model with no lesion annotations for *AI-severity* is indeed novel. We rather seek to apply state-of-the-art machine learning algorithms to tag the most at risk hospitalized COVID-19 patients. We have compared *AI-severity* to 11 published scores and some of them use modern machine learning methods to combine variables (XGBoost, deep neural network). Our evaluation of the different scores confirmed that the implemented approach for the weakly supervised approach is well-suited to construct a prognostic score from CT-scan images. We believe that this real-world example of the application of deep-learning methodologies in a global emergency context, with imperfect data and limited available annotations is a useful illustration of the power of these methodologies.

2. One previously raised question was whether the AI severity model would be able to predict clinical metadata such as age and gender only with lung features. The authors reported “AUC of 0.88 for predicting an age larger than 60 year-old”. Rather than AI could learn “to estimate variables correlated to age” from lung directly, my hypothesis is what AI learns is the natural distribution of age of the data. As the elderly older than 60 years constitute the majority of COVID-19 patients. Information about this should be addressed in the revised manuscript if possible.

Answer

We provide in supplementary table 1 the AUC scores obtained by deep learning models similar to weakly supervised model used in *AI-severity* but trained to predict patient characteristics such as age, sex or oxygen saturation. For predicting age > 60, the AUC is respectively 0.88 for KB and 0.79 for IGR, confirming that features extracted by Resnet50 and EfficientNet B0 from CT scans contain information about age.

3. The accuracy of lung lesion segmentation is relatively low in comparison to the results reported in the literature. A possible explanation could be the fact that the utilized CT slices are very pathological. It is, however, not clear if such moderate-quality segmentations can help with the following analysis such as prognosis.

Answer

Please note that the section about *AI-segment* has now been moved to a supplementary file and is not described in the main text.

The *AI-segment* model does perform very well in the task of lung vs background segmentation, as it reaches an accuracy of 99.9% when evaluated on the fully annotated scans. The distinction between sane regions and lesion regions, within the lung, presents high accuracy too (F1 score of 0.98 on fully annotated scans).

When increasing resolution and segmenting the lung lesion into 4 classes (Sane, GGO, consolidation and CP), the segmentation task is more challenging, including at the annotation level, which explains why we obtained relatively degraded performance. Encouragingly, the volumetry per lesion class does correlate well with the scores reported in the radiologist reports, which makes us confident about using segmentation in the *AI-segment* score.

4. For the evaluation of the *AI-segment* model, the authors adopted several threshold-based methods in the revised manuscript. However, there seems no validation set for this task. Also, how did the authors select the thresholds

Answer

Accuracy and F1 score were evaluated using the IGR validation cohort (it is now mentioned in the legend of supp table segmentation 1). The thresholding operation is only used when computing statistical measures of performance such as accuracy and F1-score, as the radiologist reports do not evaluate quantitatively the volume of each lesion. The retained threshold values were set empirically, to display representative information regarding the quality of *AI-segment* ability to detect presence/absence of each lesion type. However this thresholding has no impact on the estimation of volumetry and computation of the *AI-segment* score.

5. In the paper, the authors initialized the network with pretrained weights. On the other hand, the setting of the initial learning rate is 0.1 for the 3D resnet and 0.001 for the rest two networks. My concern is that these settings are so large that they may damage the pretrained model. Have the authors tried to address this issue.

Answer

The relatively large value of the initial learning rate is justified by the fact that the networks were pretrained on a simpler lung segmentation task, different from the final four-classes lesion segmentation considered here. We have controlled the ability of the networks to segment the lungs correctly all along training, showing that the pretrained model was not damaged despite the high learning rates.

6. The paper as written and structured is a bit difficult to follow throughout. Still need to be polished.

Answer

We have modified and polished the manuscript again.

Reviewers' Comments:

Reviewer #1:

Remarks to the Author:

Thanks to the authors for their rebuttal, I think the manuscript is clearer following the revisions proposed. However, I still remain concerned about the apparent disconnect with the actual tool's utility, especially the added benefit of CT-scans. Contrary to what the authors claim about the ubiquity of CT-scans in COVID-19 patients, the ACR and the STR joint statement said chest CT scans can be restricted to patients who test positive for COVID-19 and are suspected of having complications involving the lungs. This remains until now the MO of thousands of hospitals in the US and around the world. A CT-scan is still acquired based on clinical acumen, which already makes such a model rather redundant.

Since the patient numbers are low, the data imbalanced and the AUC improvement marginal (0.03), The manuscript would benefit from two things that are essential for this study:

1. A table in the main text of a confusion matrix in the two test cohorts of 150 and 135 patients that will compare the actual number of TP and FP, TN and FN of the AI-Severity model and the C&B score, so the reader knows how many of the 150 and 135 patients benefited from an improved score due to the inclusion of CT-scans and the use of the AI-Severity model, instead of a C&B score.
2. Calibration curves, along with slope and CITL measures in both the testing cohorts. The distribution of the AI-Severity score would also be beneficial (maybe added in the x-axis or as a separate figure).

Reviewer #2:

Remarks to the Author:

In this revised manuscript, Dr Blum et al respond to the comments raised by the previous reviewers. The report shows that an AI system could be used to predict COVID-19 severity with CT scans as well as clinical variables. Two questions still remain.

1. My major concern was novelty which was previously raised, as similar work has been published recently : "Liang W, Yao J, Chen A, et al. Early triage of critically ill COVID-19 patients using deep learning[J]. Nature communications, 2020, 11(1): 1-7." and "Colombi D, Bodini F C, Petrini M, et al. Well-aerated lung on admitting chest CT to predict adverse outcome in COVID-19 pneumonia[J]. Radiology, 2020: 201433." The authors are expected clarify the novelty from aspects such as clinical applicability or methodology in the manuscript.
2. The authors are suggested to report the censor points of survival Kaplan-Meier curve as it is more informative to the clinicians.

We would like to thank the reviewers for their comments and we provide our answers to their comments below.

REVIEWER COMMENTS

Reviewer #1 (Remarks to the Author):

Thanks to the authors for their rebuttal, I think the manuscript is clearer following the revisions proposed. However, I still remain concerned about the apparent disconnect with the actual tool's utility, especially the added benefit of CT-scans. Contrary to what the authors claim about the ubiquity of CT-scans in COVID-19 patients, the ACR and the STR joint statement said chest CT scans can be restricted to patients who test positive for COVID-19 and are suspected of having complications involving the lungs. This remains until now the MO of thousands of hospitals in the US and around the world. A CT-scan is still acquired based on clinical acumen, which already makes such a model rather redundant.

Our Answer

We do not recommend to use CT-scans for all patients positive for COVID19 but for hospitalized patients, which are the ones included in our study. The recommendations of several scientific societies and organizations, which read as follows, go in the same direction.

- a. ACR society recommendations: “CT should be used sparingly and reserved for hospitalized, symptomatic patients with specific clinical indications for CT. Appropriate infection control procedures should be followed before scanning subsequent patients.”
- b. Haute Autorité de santé (HAS) recommendations- or French National Authority for Health -: “Perform a chest CT scan in the event of proven respiratory symptoms requiring hospital treatment, in an rt-PCR + or suspect patient, to assess the degree of severity of the pulmonary involvement and have a baseline examination.
- c. WHO recommendations : “If available, low-dose chest CT can support the decision on regular ward admission versus ICU admission. Chest radiographs are preferred for follow-up in regular ward admission. Patients with rapid progression of COVID-19 pneumonia or diffuse lung damage need ICU admission.”

Since the patient numbers are low, the data imbalanced and the AUC improvement marginal (0.03), The manuscript would benefit from two things that are essential for this study:

1. A table in the main text of a confusion matrix in the two test cohorts of 150 and 135 patients that will compare the actual number of TP and FP, TN and FN of the AI-Severity model and the C&B score, so the reader knows how many of the 150 and 135 patients benefited from an improved score due to the inclusion of CT-scans and the use of the AI-Severity model, instead of a C&B score.

Our Answer

Following the reviewer suggestion, we now provide in the main text a confusion matrix in Figure 3. The corresponding text in the ms reads as follows “*We also computed the confusion matrix for the outcome `oxygen flow rate of 15 L/min or higher and/or the need for mechanical ventilation and/or patient death` (Figure 3). AI-severity correctly classified 3 and*

4 additional positive patients among the 44 and 40 positive patients of the development and validation cohorts when compared to C & B and 4 additional negative patients among the 106 and 95 negative patients of the cohorts.“

2. Calibration curves, along with slope and CITL measures in both the testing cohorts. The distribution of the AI-Severity score would also be beneficial (maybe added in the x-axis or as a separate figure).

Our Answer

Following the reviewer suggestion, we have computed calibration properties. Figures about calibration are provided as Supp Figure 3 and the corresponding text in the ms reads as follows. *“We also evaluated calibration properties of AI-severity using calibration plot (Supp Fig 3)²⁴. We found slope of 0.949 (0.650,1.371) (150 leftover individuals at KB) and of 0.996 (0.755,1.383) (IGR), and intercept (calibration-in-the-large) of -0.206 (-0.564,0.172) (KB) and of 0.529 (0.088,1.084) (IGR). Estimated slopes and intercepts indicated correct calibration of AI-severity for the leftover patients of the development KB cohort and an underestimation of severe outcomes for the validation IGR cohort; AI-severity predicted a mean severity of 22% (0.18,0.25) for the 135 IGR patients whereas severe outcomes occurred for 30% (0.22,0.37) of these patients“*

Reviewer #2 (Remarks to the Author):

In this revised manuscript, Dr Blum et al respond to the comments raised by the previous reviewers. The report shows that an AI system could be used to predict COVID-19 severity with CT scans as well as clinical variables. Two questions still remain.

1. My major concern was novelty which was previously raised, as similar work has been published recently : "Liang W, Yao J, Chen A, et al. Early triage of critically ill COVID-19 patients using deep learning[J]. Nature communications, 2020, 11(1): 1-7." and "Colombi D, Bodini F C, Petrini M, et al. Well-aerated lung on admitting chest CT to predict adverse outcome in COVID-19 pneumonia[J]. Radiology, 2020: 201433." The authors are expected clarify the novelty from aspects such as clinical applicability or methodology in the manuscript.

Our answer

It is true that both papers include CT-scan or X-ray to predict severity. We acknowledge that in the upper-right inset of our figure 2. However, there are several important differences between our analysis and the 2 papers mentioned.

1. Liang et al. do not use CT-scan but X-ray data.
2. Colombi et al. extracted visual and software-based quantification from CT-scan and Liang et al. used X-ray abnormalities (Yes/No). By contrast, our deep learning approaches extracted a value from CT-scan that is by construction informative about disease severity. That is a key advantage of deep learning approaches based on images. Image quantification is guided by a prespecified outcome (disease severity here) when using a supervised learning approach.
3. AI-severity has significantly better performance than the 2 other scores mentioned in 4/6 comparisons (and AUC were larger in the 6 comparisons) (see Figure 2). The deep learning model constructs a variable from CT-scan

that is guided by the severity outcome, which might explain better performance.

2. The authors are suggested to report the censor points of survival Kaplan-Meier curve as it is more informative to the clinicians.

Our Answer

We have updated Figure 1 following the reviewer suggestion.

Reviewers' Comments:

Reviewer #1:

Remarks to the Author:

Thank you to the authors for addressing my requests. While I still remain doubtful of how generalizable and portable this model is, the paper is methodologically sound. If accepted for publication, the authors need to rephrase the abstract to avoid misrepresentation of their results. The authors should remove the number of images used and keep the number of patients used in this study. Also, since the authors have not provided any solid proof (external validation of large numbers of patients) that this is either generalizable, portable, or significantly better than the >300 COVID-19 severity scores available right now, they need to rephrase the last two sentences. I suggest the following:

"When comparing AI-severity with 11 existing scores for severity, we find significantly improved prognosis performance in our validation dataset of 285 patients; Our results suggest that AI-severity can become a useful severity scoring approach for COVID-19 patients."

We would like to thank the reviewer 1 for his comments. Please find his comment and our answer below.

Comment of reviewer 1

Reviewer #1 (Remarks to the Author):

Thank you to the authors for addressing my requests. While I still remain doubtful of how generalizable and portable this model is, the paper is methodologically sound. If accepted for publication, the authors need to rephrase the abstract to avoid misrepresentation of their results. The authors should remove the number of images used and keep the number of patients used in this study. Also, since the authors have not provided any solid proof (external validation of large numbers of patients) that this is either generalizable, portable, or significantly better than the >300 COVID-19 severity scores available right now, they need to rephrase the last two sentences. I suggest the following:

"When comparing AI-severity with 11 existing scores for severity, we find significantly improved prognosis performance in our validation dataset of 285 patients; Our results suggest that AI-severity can become a useful severity scoring approach for COVID-19 patients."

Our answer

Following the reviewer suggestion, we have removed in the abstract the counting of the number of images. In addition to this, the manuscript now ends with the following sentences "When comparing AI-severity with 11 existing scores for severity, we find significantly improved prognosis performance in the validation datasets of 150 and 135 patients. Our results suggest that AI-severity can become a useful severity scoring approach for COVID-19 patients."